# Dissection of amino acid acquisition pathways demonstrates that amino acid starvation of *Borrelia burgdorferi* results in a (p)ppGpp-independent maladaptive response

Arti Kataria[1], Eric Bohrnsen[2], Benjamin Schwarz [ORCID][2], Dan Drecktrah[3], D. Scott Samuels [ORCID][3], Aaron B. Carmody [ORCID][4], Lara M. Myers [ORCID][4] & Ashley M. Groshong [ORCID][1] ✉

*Borrelia burgdorferi*, the causative agent of Lyme disease, is well known for its unique physiology and enzootic cycle. Building on previous work showing peptide transport is essential for viability, we endeavored to clearly define the impact of peptide starvation on the spirochete and directly compare peptide starvation to targeted free amino acid starvation. Herein, we confirm the ability of a putative glutamate transporter, BB0401, to transport glutamate and aspartate as well as demonstrate its requirement for viability. Using conditional mutants for both peptide transport and BB0401, we characterize these systems throughout the enzootic cycle, confirming their essential role during murine infection and revealing that they are dispensable during prolonged colonization of the tick midgut. We broadly define the metabolic perturbations resulting from these starvation models and show, even under the most severe amino acid stress, *B. burgdorferi* is unable to modulate its physiological response via the canonical (p)ppGpp-driven stringent response.

Bacterial genomes remain in an evolutionary flux, adjusting to provide physiological support for bacteria to survive within their established niches or to adapt to new ones[1]. Non-fastidious bacteria often retain much of their biosynthetic capabilities, allowing them to survive diverse nutritional environments. Alternatively, fastidious bacteria may relinquish synthetic pathways to allow for pathogenic adaptations or to improve overall cellular economy. The *Borrelia burgdorferi* genome is well-tailored for its enzootic lifestyle. Several metabolic synthesis pathways have been abandoned given its predictable microenvironments, and genomic real estate has been allocated to its multitude of endogenous plasmids, which contain numerous paralogous surface proteins that define and engage with its narrow vector and broad host ranges[2,3]. The Lyme disease spirochete encodes very few regulatory systems[4–8], primarily relying on the cyclic di-nucleotide second messenger c-di-guanosine monophosphate

(c-di-GMP) controlling tick-phase gene regulation and the RNA polymerase sigma N/RNA polymerase sigma S (RpoN/RpoS; BB0450/BB0771) sigma factor cascade for vertebrate-phase gene regulation. *B. burgdorferi* has been shown to utilize the alarmone 3',5'-bis(pyrophosphate)- and guanosine pentaphosphate- ([p]ppGpp) to induced the stringent response under starvation conditions[9–11]. The spirochete encodes a RelA/SpoT homolog (Rel$_{Bbu}$; BB0198) with dual functionality to synthesize and hydrolyze (p)ppGpp[9]. Additionally, *B. burgdorferi* encodes a DnaK suppressor protein (DksA; BB0168), which is a transcription factor through which (p)ppGpp affects transcription in response to nutrient flux[12,13]. The stringent response in the spirochete has been shown to be critical for survival during the transition from the vegetative cell state in the nutrient-replete molted tick and to the subsequent blood meal[9,13].

[1]Laboratory of Bacteriology, Rocky Mountain Laboratories, Division of Intramural Research, National Institute of Allergy and Infectious Diseases, National Institutes of Health, Hamilton, MT, USA. [2]Proteins and Chemistry Section, Research Technologies Branch, Rocky Mountain Laboratories, Division of Intramural Research, National Institute of Allergy and Infectious Diseases, National Institutes of Health, Hamilton, MT, USA. [3]Division of Biological and Biomedical Sciences, University of Montana, Missoula, MT, USA. [4]Flow Cytometry Section, Research Technologies Branch, Rocky Mountain Laboratories, Division of Intramural Research, National Institute of Allergy and Infectious Diseases, National Institutes of Health, Hamilton, MT, USA. ✉e-mail: ashley.groshong@nih.gov

Amino acids are crucial building blocks, needed not only for peptide synthesis but also nitrogen and carbon metabolism, cell-cell communication, and peptidoglycan synthesis; they also serve as intermediates for many metabolic pathways[14]. Canonical bacterial responses to amino acid starvation are hardwired into their transcriptional response via (p)ppGpp and the stringent response[12]. Upon sensing amino acid limitations, bacteria are able to reduce cell growth and/or depend on biosynthetic pathways to support essential amino acid requirements. In the absence of amino acid synthesis pathways, *B. burgdorferi* depends on transport of peptides or free amino acids to maintain its intracellular free amino acid pools[15–19]. We had previously characterized a conditional mutant of the oligopeptide transport system (Opp) and demonstrated that *B. burgdorferi* depends on peptide transport for viability[19]. Additionally, we were able to demonstrate that there was only a small window (~48 h) in which starved spirochetes could recover and prolonged starvation resulted in cell elongation, which was unexpected given the limitation of core building blocks. To refine our understanding of amino acid starvation in *B. burgdorferi*, we sought to compare our peptide starvation model with that of a targeted amino acid deficiency. The partially defined and complex nature of *B. burgdorferi* cultivation media prevents direct media modifications for something as ubiquitous as amino acids[20]. Therefore, as with peptide transport, we sought to target a specific amino acid transporter.

The Glt transporters are part of a larger family of dicarboxylate/amino acid:cation symporters (DAACS)[21]. These transporters are known to symport sodium ions and/or protons along with their designated substrate(s). The Glt transporters are a subgroup of DAACS that symport L-glutamate, although many have been shown to transport aspartate either preferentially or in tandem. *E. coli* encodes both a GltP (GltP$_{Ec}$) and GltS (GltS$_{Ec}$), and these transporters demonstrate differences in ion and substrate selectivity[22,23]. GltP$_{Ec}$ facilitates proton-dependent symport of L-glutamate, L-aspartate, and D-glutamate while GltS$_{Ec}$ is sodium-dependent and is primarily an L-glutamate symporter with some tolerance for D-glutamate and L-glutamine[22–24]. The two GltP homologs with solved crystal structures are from archaea, *Pyrococcus horikoshii* (Glt$_{Ph}$) and *Thermococcus kodakarensis* (Glt$_{Tk}$)[25–27]. Unlike GltP$_{Ec}$, Glt$_{Ph}$ has been shown to utilize sodium during symport; however, there are conflicting reports about its utilization of protons[26,28,29]; Glt$_{Tk}$ appears to exclusively use sodium ions[27]. Additionally, both transporters have been shown to exclusively transport L- and D-aspartate instead of glutamate[26,27,29]. Other Glt homologs have been characterized with respect to transport selectivity, with *Bacillus stearothermophilus* GltT (GltT$_{Bs}$) similar to GltP$_{Ec}$ in transport of L-glutamate, L-aspartate, and D-glutamate while *Bacillus subtilis* GltP (GltP$_{Bs}$) was shown to primarily transport L-glutamate and L-aspartate with minimal transport of D-glutamate, L-glutamine, and L-asparagine[23,30].

Herein, we characterize the putative GltP (BB0401) in *B. burgdorferi*[3,31]. Structurally, Swiss-Model[32] generated BB0401 models using Glt$_{Ph}$ (PDB ID 2nwl)[26] as a template fit well with other GltP homologs, though binding residues are not highly conserved. We compare a conditional mutant for *bb0401* to our peptide starvation model using the conditional Opp transport mutant, allowing us to evaluate the impact of a targeted, limited amino acid starvation to that of a broad, severe amino acid starvation. The phenotypes of the two mutants appear somewhat similar, with the loss of *bb0401* affecting growth and morphology as with the Opp system; however, we show that the impacts to cellular processes are distinct. We characterize both the *bb0401* and *opp* conditional mutants in the laboratory model of the enzootic cycle using in vivo IPTG-supplementation, which represents the first use of the IPTG-inducible system for studies of *B. burgdorferi* in ticks and exposes an unexpected capacity for these mutants to survive within the tick vector given their growth phenotypes in vivo. Metabolomics comparisons define the breadth of the metabolic lesions incurred by these starvation models and heavy labeling experiments confirm that BB0401 is indeed a functional GltP. Finally, we use our peptide starvation model to demonstrate that *B. burgdorferi* has uncoupled its amino acid starvation response from the (p)ppGpp-driven stringent response, which accounts for the maladaptive phenotype observed in the *opp* conditional mutant.

## Results

### BB0401 is the only encoded GltP homolog but demonstrates minimal binding site conservation

Prior annotations of the *B. burgdorferi* genome originally categorized BB0729 as a GltP homolog, with later analyses uncovering BB0401 as an additional homolog[31]. However, Eggers et al. posited that BB0729 was likely a Na + /cystine symporter, TcyP, based on high homology to the *B. subtilis* TcyP; this inference was supported by its co-transcription with *bb0728*, the *B. burgdorferi cdr* gene encoding CoA-disulfide reductase, as L-cystine is required to generate CoA[33]. The human cystine transporter SLC7A11 is a cationic amino acid transporter (CAT) and was included as an outgroup. To evaluate the relationships between BB0401, BB0729, and other DAACs homologs, we generated a phylogenetic tree with homologs from representative human and bacterial species (Fig. S1). Notably, BB0401 and other spirochete homologs (labeled as GltP) diverged early from other DAACs and are separate from bacterial and archaeal GltPs. Meanwhile, BB0729 and other *Borrelia* homologs (labeled TcyP) make a distinct clade with other TcyP homologs. While BB0401 is more distantly related to other bacterial DAACs, BB0729 specifically clusters with the TcyP homologs, strengthening the evidence that it may not serve as *B. burgdorferi*'s GltP homolog.

Glt$_{Ph}$ (PDB ID 2nwl)[26] has been shown to form a homotrimer (Fig. 1a) consistent with the closely related eukaryotic excitatory amino acid transporters (EAATs) operative in the neuronal glutamate circuit[24–27,34]. Each protomer can transport cargo and trimerization is believed to provide stability within the membrane. The Glt$_{Ph}$ binding pocket for aspartate consists of nine liganding amino acids[24] (Figs. 1b-d and h, S2b). Of note, four of the nine aspartate-binding residues interact via main-chain carbonyl groups instead of side groups. Binding of sodium ions in Glt$_{Ph}$ (PDB ID 2nwx) have been described for two sites adjacent to the aspartate binding pocket and consists of five amino acids for site 1 and four amino acids for site 2[26] (Figs. 1d and h, S2d). In a previous study, homology modeling of GltP$_{Ec}$ against the structure of Glt$_{Ph}$ demonstrated a high degree of conservation among the aspartate binding site, though docking studies suggested the glutamate binding site was slightly shifted from the predicted aspartate binding site[24] (Fig. 1c). Rahman et al. also demonstrated that accessory residues T314 and M362 in Glt$_{Ph}$ influence cargo specificity as mutagenesis of these residues increased glutamate binding[24]. The same study demonstrated that an L313M mutation in GltP$_{Ec}$ ablates glutamate transport, suggesting this substitution in Glt$_{Ph}$ (M311) may contribute to loss of L-glutamate transport[24].

We generated BB0401 and GltP$_{Ec}$ homology models using SwissModel[32] based on the crystal structure of Glt$_{Ph}$ (PDB ID 2nwl). The GMQE value for the BB0401 model was 0.57 and QMEANDisCo value was 0.54 ± 0.05. The GMQE value for the GltP$_{Ec}$ model was 0.64 and QMEANDisCo value was 0.64 ± 0.05. Given the low resolution of 2nwl and the low homology between BB0401 and Glt$_{Ph}$, we also modeled BB0401 as a homotrimer using AlphaFold2-Multimer[35]; the AlphaFold model aligned well with the model generated by SwissModel (Fig. S2a, RMSD 1.431 Å). Given the similarity between the two models, we proceeded with the BB0401 model based on the 2nwl crystal structure for simplicity. Using the model of BB0401 we compared overall structure (Fig. 1a) and binding sites (Fig. S2b-d) with that of Glt$_{Ph}$[26] and the predicted sites for GltP$_{Ec}$[24]. Both models generated by SwissModel model for BB0401 and GltP$_{Ec}$ aligned well with the Glt$_{Ph}$ structure and the RMSD value of Glt$_{Ph}$ and BB0401 was 0.587 Å and Glt$_{Ph}$ and GltP$_{Ec}$ was 0.303 Å. (Fig. 1a). While liganding residues showed consistent positional alignments within the different binding sites (Fig. S2b-d), unlike GltP$_{Ec}$, BB0401 residues lining the different binding pockets showed limited conservation, specifically with respect to amino acid binding sites (Fig. 1h). However, of the six non-conserved residues in the aspartate binding site of BB0401, three are residues that interact via the main-chain carbonyl group (Figs. 1a and h and S2b). Of note, key residue D405 in sodium binding site 1 of Glt$_{Ph}$ correlates with N401 in GltP$_{Ec}$ and N381 in BB0401. A D405N Glt$_{Ph}$ mutant resulted in loss of sodium binding at Site 1[26], consistent with reports that GltP$_{Ec}$ is not sodium dependent, and suggests that BB0401 may behave similarly. Given the lack of binding site

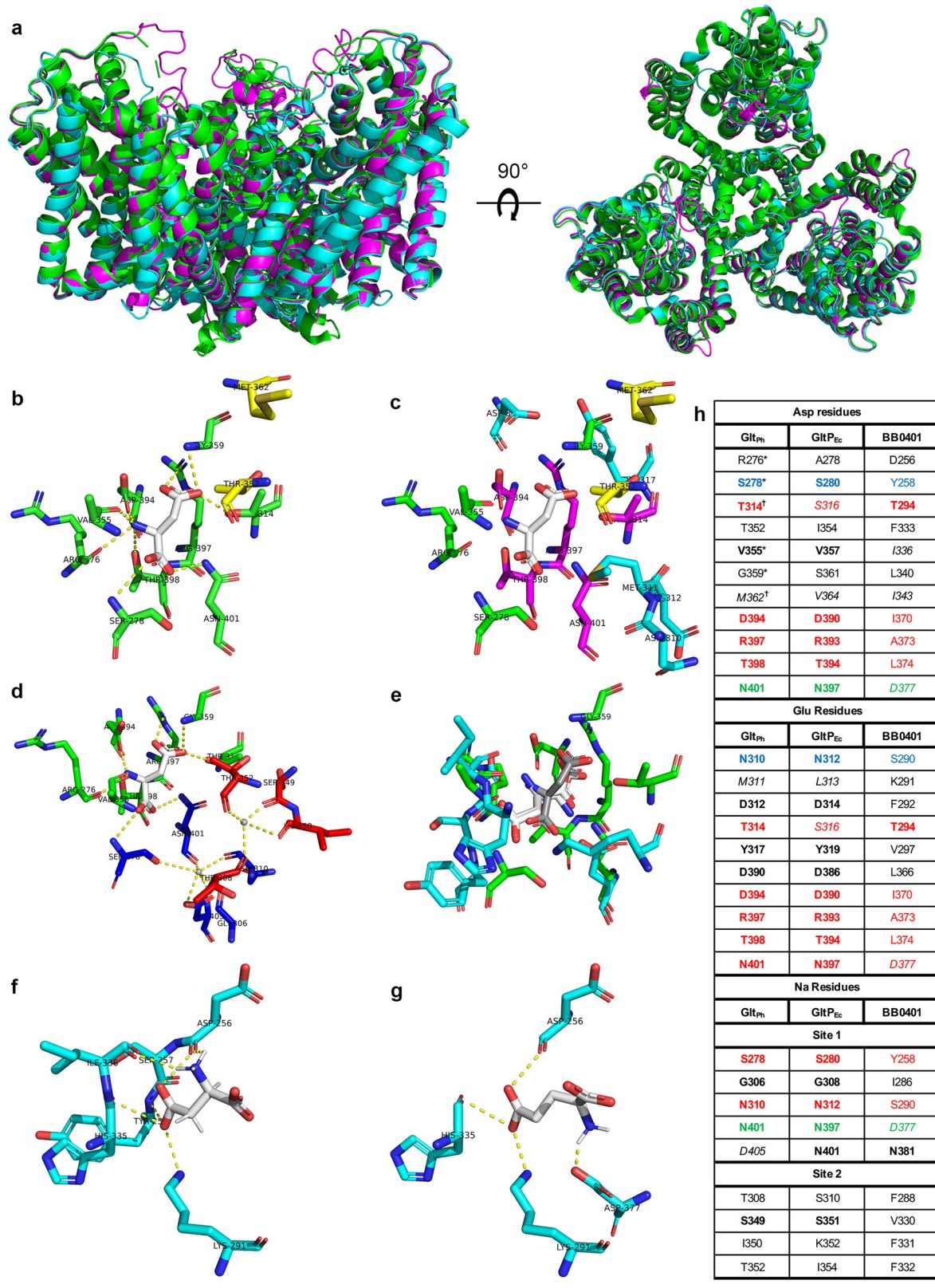

| Asp residues | | |
| --- | --- | --- |
| Glt$_{Ph}$ | GltP$_{Ec}$ | BB0401 |
| R276* | A278 | D256 |
| S278* | S280 | Y258 |
| T314† | S316 | T294 |
| T352 | I354 | F333 |
| V355* | V357 | I336 |
| G359* | S361 | L340 |
| M362† | V364 | I343 |
| D394 | D390 | I370 |
| R397 | R393 | A373 |
| T398 | T394 | L374 |
| N401 | N397 | D377 |

| Glu Residues | | |
| --- | --- | --- |
| Glt$_{Ph}$ | GltP$_{Ec}$ | BB0401 |
| N310 | N312 | S290 |
| M311 | L313 | K291 |
| D312 | D314 | F292 |
| T314 | S316 | T294 |
| Y317 | Y319 | V297 |
| D390 | D386 | L366 |
| D394 | D390 | I370 |
| R397 | R393 | A373 |
| T398 | T394 | L374 |
| N401 | N397 | D377 |

| Na Residues | | |
| --- | --- | --- |
| Glt$_{Ph}$ | GltP$_{Ec}$ | BB0401 |
| Site 1 | | |
| S278 | S280 | Y258 |
| G306 | G308 | I286 |
| N310 | N312 | S290 |
| N401 | N397 | D377 |
| D405 | N401 | N381 |
| Site 2 | | |
| T308 | S310 | F288 |
| S349 | S351 | V330 |
| I350 | K352 | F331 |
| T352 | I354 | F332 |

conservation, we used SwissDock to simulate binding of aspartate and glutamate in our model of BB0401. We found that both amino acids were predicted to bind within the same aspartate binding pocket identified in Glt$_{Ph}$ (Fig. 1e). Hydrogen bond predictions in PyMOL suggested only partial conservation with the Glt$_{Ph}$ aspartate liganding residues (BB0401 residues

D256/I336 for Asp and D256/D377 for Glu) and showed no conservation with the glutamate binding site predicted for GltP$_{Ec}$[24] with the exception of D377 which is shared with Glt$_{Ph}$[26] aspartate binding (Fig. 1f-g). The disparity between binding pocket residues for BB0401 and GltP$_{Ec}$/Glt$_{Ph}$ suggest differences in substrate specificity or transport mechanisms for BB0401.

**Fig. 1 | Residues lining the predicted BB0401 binding pocket are not highly conserved. a** Side and top view of Glt$_{Ph}$ (2nwl; green), GltP$_{Ec}$ homology model (magenta), and BB0401 homology model (cyan) alignment. **b** Residues involved in Asp binding in crystal structure of Glt$_{Ph}$ complexed with Asp (PDB ID 2nwl), Asp is shown in white, accessory residues determined by Rahman et al., in yellow, bonds denoted with yellow dashed lines. **c** Comparison of Asp binding site shown in the crystal structure of Glt$_{Ph}$ complexed with Asp (PDB ID 2nwl) and corresponding residues as predicted Glu binding site as determined by AutoDock in Rahman et al. Asp is shown in white, residues unique to Asp binding in green, residues unique to Glu binding in cyan, and residues which overlap the two binding sites in magenta. Accessory residues identified by Rahman et al. are shown in yellow. **d** Residues involved in Na binding in crystal structure of GltPh complexed with Asp and Na (PDB ID 2nwx). Asp is shown in white, Asp liganding residues in green, Na binding site 1 in blue, Na binding site 2 in red, and bonds denoted with yellow dashed lines. **e** Overlay of Asp binding site from crystal structure of Glt$_{Ph}$ complexed with Asp (PDB ID 2nwl) and predicted Asp and Glu binding sites in the BB0401 homology model as determined by SwissDock. Asp from the GltPh crystal structure is shown in grey, Asp and Glu from BB0401 SwissDock both shown in white, residues for Glt$_{Ph}$ binding in green, and putative residues for BB0401 binding as determined by SwissDock in cyan. Individual models of BB0401 SwissDock binding shown for (**f**) Asp and (**g**) Glu. Ligands shown in white, bonds denoted as yellow dashed lines. Structures were visualized in PyMol. h) Residue table showing corresponding amino acids based on the Asp liganding residues in Glt$_{Ph}$ crystal structure complexed with Asp (PDB ID 2nwl), the Glu binding residues for GltP$_{Ec}$ as determined by AutoDock in Rahman et al., and the two Na binding site residues as defined in Glt$_{Ph}$ crystal structure complexed with Asp and Na (PDB ID 2nwx). Identical amino acids are in bold, similar amino acids as defined by BLOSUM62 are in italics. Residues that are shared between amino acid binding sites are in red, residues shared with Na binding sites are in blue, and residues shared among all three sites are in green. *Glt$_{Ph}$ aspartate binding residues with main-chain carbonyl group interactions. †Residues identified by Rahman et al. which can increase binding of Glu when mutated in Glt$_{Ph}$.

## *bb0401* is important for growth in vitro

As previously shown, peptide transport is required for growth as demonstrated via an *opp*$^{cond}$ mutant[19]. The *opp*$^{cond}$ mutant was generated by targeting the nucleotide binding domain containing proteins (OppDF; BB0334-5), thus allowing abrogation of peptide transport from all five binding proteins in a single mutant. To characterize the contributions of BB0401 to *B. burgdorferi* growth and the enzootic cycle, we set out to construct a *bb0401* mutant. We generated a Δ*bb0401* suicide vector (pΔ*bb0401*, Table S1, Fig. S3a) that replaces the *bb0401* open reading frame (ORF) with a gentamicin antibiotic selection marker (P$_{flgB}$-*aacC1*) via homologous recombination. Repeated attempts to transform our wild-type (*wt*) strain (Table S1) with pΔ*bb0401* failed, suggesting the gene may be essential for in vitro cultivation. Therefore, we utilized the conditional mutagenesis approach that had previously been used to generate *opp*$^{cond}$[19]. A shuttle vector containing an IPTG-inducible *bb0401*, constitutive *lacI*, and P$_{flgB}$-*aadA* streptomycin antibiotic selection marker (p*ibb0401*, Table S1, Fig. S3b) was generated. p*ibb0401* was transformed into *wt*, and a single streptomycin resistant clone was selected for subsequent transformation with pΔ*bb0401* in the presence of 1 mM IPTG. A single clone (*bb0401*$^{cond}$) was confirmed to have the *bb0401* ORF replaced with the P$_{flgB}$-*aacC1* selectable marker and carry the inducible shuttle vector as well as all parental plasmids except cp32-3, cp9, and lp5, which are plasmids known to be dispensable for the enzootic cycle[36] (Fig. S3c).

We sought to quantify the changes in expression of *bb0401* due to IPTG titration within this system. While analysis of protein expression is ideal in such a system, and has indeed been used to confirm expression for previous conditional mutants[37–44], attempts to express either full length or truncated BB0401 failed. Additionally, generation of serum against a synthesized peptide from BB0401 failed to detect the protein in *B. burgdorferi* whole cell lysates. Instead, we turned to a semi-quantitative analysis of full-length transcripts using RT-PCR (Fig. 2a). We tested 10-fold reductions in IPTG concentrations to assess the changes in transcription levels of *bb0401* compared to the housekeeping gene *flaB*. We performed first strand cDNA reactions on equivalent amounts of RNA and performed subsequent PCR with our full-length primers to visualize the relative quantities of transcript for our targets. We found a significant reduction of intact *flaB* transcripts in cells treated with 0.01 mM or no IPTG, suggesting RNA degradation in starved cultures. 1 mM and 0.1 mM IPTG concentrations appeared to have higher levels of transcript compared with the *wt* strain. Alternatively, 0.01 mM IPTG and absence of IPTG showed to detectable *bb0401* transcript as expected under starvation conditions. We noticed a difference in size from our genomic DNA positive control and our RT-PCR samples for *bb0401*, though the *wt* and *bb0401*$^{cond}$ transcripts were of the same size. To confirm this size difference was not due to some modification of the transcript, we sequenced the PCR products and found no changes to the transcript sequence. We believe the size difference is an artifact of the large increase in the loading volumes required to visualize the RT-PCR bands for *bb0401* compared to the minimal load of the diluted sample of gDNA for the positive control.

To evaluate the loss of *bb0401* expression on growth we performed a growth curve using titrations of inducer for *bb0401*$^{cond}$ (Fig. 2b) and found that without inducer, *bb0401*$^{cond}$ demonstrated a significant defect in growth. With respect to IPTG concentrations, there was a more narrow window for intermediate growth (Fig. 2b, 0.02–0.03 mM) compared with *opp*$^{cond}$ (0.04–0.08 mM)[19] as well as a lower overall threshold for optimal growth (Fig. 2b, 0.04 mM), as *opp*$^{cond}$ required at least 0.1 mM IPTG to attain normal growth[19]. We also noted that *bb0401*$^{cond}$ was able to proliferate during the first three days of cultivation without IPTG (~100–1000-fold increase overall; Fig. 2b) in contrast to *opp*$^{cond}$, which demonstrated no growth once inducer was removed from culture[19].

## *bb0401*$^{cond}$ in vitro phenotype is distinct from *opp*$^{cond}$

Abrogation of the Opp system and subsequent peptide starvation in *B. burgdorferi* resulted in unexpected phenotypes related to growth and morphology[19]. Therefore, we sought to compare the broad amino acid starvation phenotype of the Opp system with the more focused starvation of *bb0401*$^{cond}$. Previously, we had shown that *opp*$^{cond}$ was unable to recover growth after approximately 24-48 h of "starvation" (e.g., without inducer)[19]. Surprisingly, *bb0401*$^{cond}$ was able to recover from lack of inducer over a significant period of starvation (Fig. 2c, d). Initially, we performed a recovery curve where aliquots of *bb0401*$^{cond}$ with no inducer were spiked with 1 mM IPTG over d 0-7; however, we only noted a small defect in recovery after seven days of starvation (Fig. 2c). Therefore, we performed an additional recovery curve over d 6-13 (Fig. 2d). While we observed similar recovery curves as seen with *opp*$^{cond}$, where peak densities started to drop with successive recovery curves after a critical period of starvation[19], *bb0401*$^{cond}$ was able to starve significantly longer than *opp*$^{cond}$ without impacting recovery (d 6-7 vs d 1-2, respectively). Overall, growth curves and recovery curves between the two mutants demonstrated similar trends with differences in initial growth before stalling and ability to recovery indicative of either a severe or modest starvation in *opp*$^{cond}$ and *bb0401*$^{cond}$, respectively.

Given the drastic difference in growth phenotypes between the two mutants, we sought to quantify cell viability during starvation. For a global assessment of cell viability, we utilized live/dead staining coupled with flow cytometry, comparing *opp*$^{cond}$ and *bb0401*$^{cond}$ survival without IPTG to *wt* grown at 37 °C (Fig. 2e). Gating strategies to identify live/dead populations are shown in Fig. S4a-b. To minimize mutant perturbations during centrifugations, flow was performed on samples in BSK-II media. Syto9$^+$/PI$^-$ and Syto9$^+$/PI$^+$ defined our live and dead cells, respectively. *Wt* was able to maintain ~80% viability up to seven days, approximately 2 d after reaching stationary phase, then dropped to ~20%. Consistent with the published recovery curve[19], *opp*$^{cond}$ viability dropped to ~30% by d 3, the point at which normal growth could not be recovered. A low level of viability was maintained by *opp*$^{cond}$ until d 13-14 where viability is near zero. In contrast, and consistent with recovery curves in Fig. 2c, d, *bb0401*$^{cond}$ viability drops to

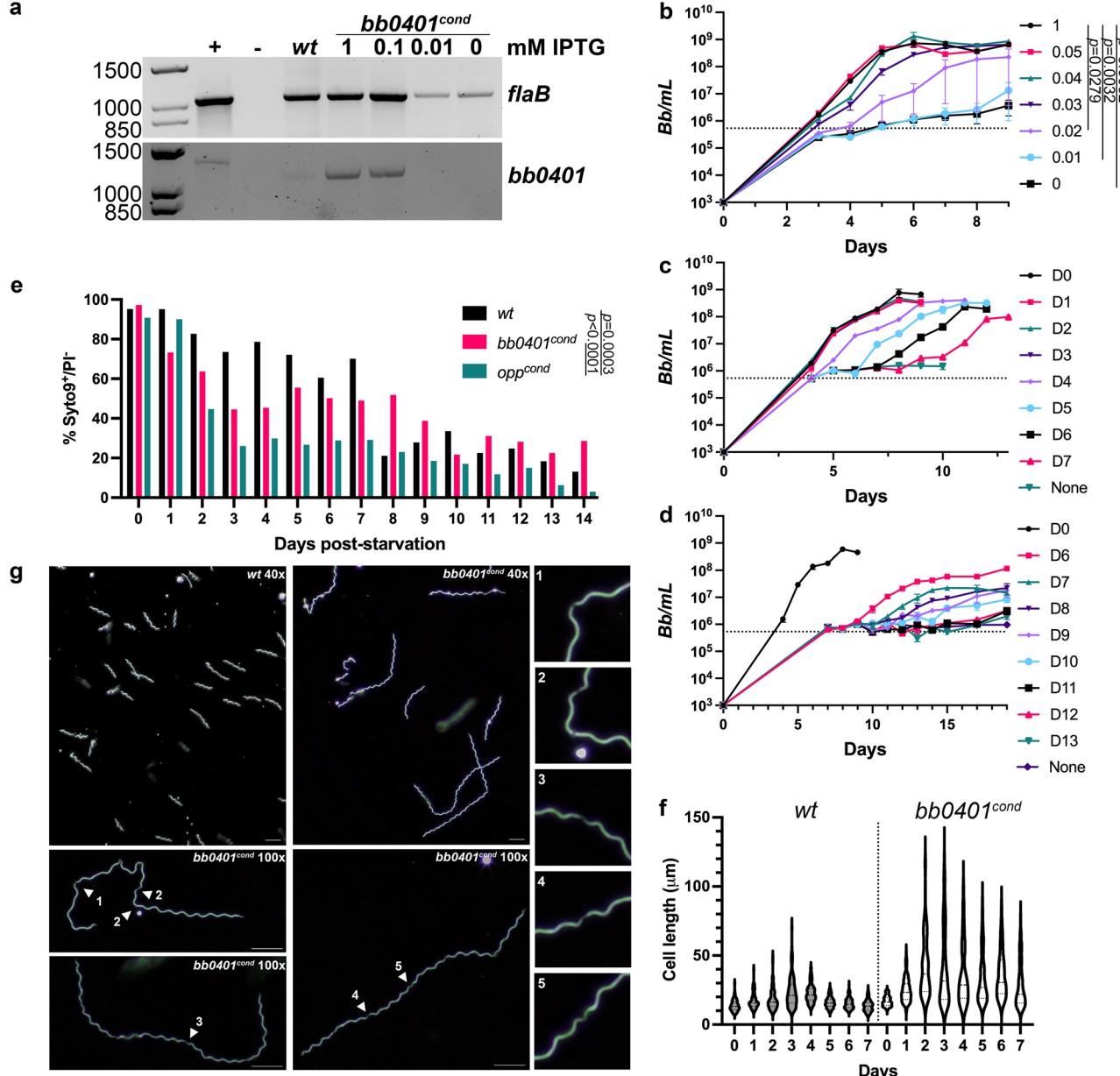

**Fig. 2 | Starvation of *bb0401^cond* significantly impacts growth. a** RT-PCR amplification of full transcripts for *bb0401* and *flaB* using equivalent amounts of RNA. "+" represents 20 ng gDNA for *flaB* and 0.2 ng gDNA for *bb0401*. Loading volumes for *bb0401* RET-PCR samples were 15x greater than *flaB* to allow for transcript visualization. **b** Growth curve demonstrating the effects of IPTG titration (0-1 mM) on *bb0401^cond* at an inoculation of 10³ spirochetes/ml. Error bars represent ±SEM, *n* = 3, *p* values were determined by two-way ANOVA. Graph is representative of three independent growth curves. **c, d** Recovery growth curves of *bb0401^cond* demonstrating the ability of the starved culture to grow with the addition of 1 mM IPTG on the days designated in the key. Dotted lines denote the limit of detection by darkfield microscopy. Error bars represent ±SEM, n = 3. Graphs are representative of three independent recovery curves. **e** Percent viability as determined by flow cytometry of *wt*, *bb0401^cond*, and *opp^cond* without IPTG at 37 °C. *p* values were determined by two-way ANOVA. **f** Violin plot showing spirochete length for *wt* and *bb0401^cond* without IPTG as measured by ImageJ. D0 represents normal cell length before starvation. Length measurements for *wt* and *bb0401^cond* (*n* > 100) are compared by two-way ANOVA, *p* < 0.0001. **g** Darkfield microscopy showing *wt* and *bb0401^cond* without IPTG for 3 days with a 40x or 100x objective. Scale bars represent 10 µm. Numbered arrows show to septation points in cell body and correlate with numbered panels below.

~50% over d 3-8 and ultimately hovers near ~30% by d 14. *B. burgdorferi* is frequently grown at room temperature to simulate incubation within an unfed tick, a state in which spirochetes undergo little proliferation[9,45]. To determine how reduced pressure to proliferate impacts these mutants, we conducted flow cytometry on cultures incubated at room temperature (Fig. S4c). All cultures maintained >60% viability over a four-week period, the approximate time required for fed larvae to molt.

An unexpected phenotype observed during starvation of *opp^cond* was cell elongation[19]. When grown without IPTG, *opp^cond* was observed to produce a heterogenous population of short and elongated cells, and cell

elongation progressed up to about tenfold normal cell length over time. We also observed heterogenous cell length populations when *bb0401^cond* was grown without IPTG (Fig. 2f, g). As with *opp^cond*, many of the longest cells were unable to be measured as they create tangled masses in which individual cells cannot be distinguished; however, we were able to clearly show cell elongation for *bb0401^cond* under starved conditions with similar cell length heterogeneity as seen with *opp^cond*. Interestingly, there are some clear differences in elongated cell morphology compared with *opp^cond*. Elongated *opp^cond* often had extensive regions of cell flattening at the cell center that was shown to be due to loss of flagellar overlap, and cell septation could not be

visualized in elongated spirochetes[19]. In contrast, *bb0401^{cond}* elongated cells maintain planar waveform, and cell septation can be seen at regular intervals (Fig. 2g). Additionally, *opp^{cond}* maintained motility in elongated cells, with cells that demonstrated cell flattening still motile at cell poles where flagella were intact[19]. Unexpectedly, *bb0401^{cond}* showed little to no motility after 2 d without inducer, with only occasional twitching despite preservation of planar waveform. These data suggest that while the starvation conditions may be superficially similar, there are distinct phenotypes elicited by targeted free amino acid starvation and peptide starvation.

## *bb0401* and *opp* are crucial for murine infection

We sought to determine whether one or both transport systems are critical for the murine model of infection. Previously, we attempted to utilize an IPTG supplementation in vivo model to test infectivity of *opp^{cond}*, which resulted in serological conversion when mice were needle-inoculated and fed IPTG-water but no culture-positive tissues were recovered[19]. We subsequently found that the original *opp^{cond}* mutant used in the study had spontaneously lost lp28-1 during cultivation, a plasmid that is essential for infection[36] (Fig. S3c). Therefore, for this infection study we selected a different *opp^{cond+28-1}* clone what was missing lp21, a plasmid not known to have an impact on the enzootic cycle[36], but contained lp28-1 (Fig. S3c). We intradermally needle-inoculated mice with $10^4$ spirochetes of strains *wt*, *bb0401^{cond}*, or *opp^{cond+28-1}*; mutants were inoculated into two different cohorts that were either given water or water supplemented with IPTG. To evaluate infections, we collected sera for western blotting (Fig. 3a, b) and collected tissues for culture (Table 1) and qPCR (Fig. 3c, d) at 2 wk post-inoculation. In contrast to *wt*, we found that all mice inoculated with *opp^{cond+28-1}* without IPTG-supplemented water were seronegative (Fig. 3a-b) and culture negative (Table 1) at 2 wk post-inoculation. While *bb0401^{cond}* showed some seroreactivity in mice without IPTG, the antibody response was less robust and likely indicative of a short-term survival before being cleared, consistent with the mutant's ability to survive in vitro starvation for longer periods of time. Alternatively, all mice that received IPTG-water were culture positive for all tissues from mice infected with either *bb0401^{cond}* or *opp^{cond+28-1}* (Table 1). qPCR of tissues at 2 wk post-inoculation (Fig. 3c-d) demonstrated burdens consistent with *wt* for mice that received IPTG; the only statistically significant difference was an approximate tenfold reduction in heart tissue for *opp^{cond+28-1}* (Fig. 3d). Alternatively, mice that did not receive IPTG showed no detectable burdens by qPCR. IPTG-supplementation was maintained until 4 wk post-inoculation in a subset of mice used to perform larval acquisition studies and mice maintained positive tissue cultures even at this later timepoint (Table 1). We were able to detect transcripts of *bb0401* in hearts infected with *wt* or *bb0401^{cond}* where mice received IPTG (Fig. S5c) after 4 wk post-inoculation; transcripts of the mutant were consistent with *wt* and confirm that IPTG levels in the tissue were sufficient to induce normal levels of transcript. Overall, these data confirm that both *bb0401* and *opp* are essential for murine infectivity. Gilbert et al.[46] used an in vivo inducible gene system to control OspC expression, a critical gene for early colonization, in mice. This study demonstrated complementation via IPTG-induction of OspC in an OspC mutant at 1 wk post-inoculation, though not all tissues were culture positive[46]. Here, we see the ability of the IPTG-complementation model to maintain infection for up to 4 weeks post-inoculation with viable spirochetes recovered from all tissues. Because IPTG-inducible systems have been shown to accrue mutations in the *lac* operator when expression is essential for growth[41], we confirmed that the mutant's ability to survive in tissues was not due to loss of IPTG sensitivity by passaging cultures from each tissue into media with or without IPTG to test IPTG sensitivity.

## *bb0401* and *opp* are not required for persistence within the molting tick

To determine how important these transport systems are for tick colonization, we fed naïve *Ixodes scapularis* larvae on mice infected with *wt* and mice infected with *bb0401^{cond}* or *opp^{cond+28-1}* that were receiving IPTG-supplementation at 2 wk post-inoculation. At drop-off, fed larvae

were pooled and evaluated for burdens via semi-solid plating and qPCR (Figs. 3e, f and S5a-b; FL DO). Given that IPTG should be present in the bloodmeal of larvae fed on mutant-infected mice supplemented with IPTG, we expected that the larvae would be successfully colonized. Indeed, both *bb0401^{cond}* and *opp^{cond+28-1}* colonized larvae similar to *wt* as shown by plating and qPCR. To determine whether subsequent digestion of the bloodmeal and tick molting, a process expected to deplete residual IPTG, would impact colonization with the mutant strains, pools of larvae were collected every 2 wk post-drop off until larvae molted (Figs. 3e, f and S5a-b; 2-6wk PDO for *bb0401^{cond}* and 2-4 wk PDO for *opp^{cond+28-1}*) and after molt (Figs. 3e, f and S5a-b; MN) for evaluation by semi-solid plating and qPCR. Surprisingly, *bb0401^{cond}* and *opp^{cond+28-1}* survived as well as *wt* during the transition from fed larvae to flat nymph. Ten colonies were isolated from each pool of ticks used in plating and confirmed to be IPTG responsive during in vitro culture.

## *bb0401* and *opp* are required for survival within the feeding nymph and transmission

Finally, we evaluated the ability of *bb0401^{cond}* and *opp^{cond+28-1}* to replicate within the feeding nymph and transmit to naïve mice. We confirmed colonization of molted nymphs 1 d prior to transmission experiments (Figs. 3e, f and S5a-b; MN PF), which was approximately one month after molt. Ten nymphs colonized with *wt, bb0401^{cond}*, or *opp^{cond+28-1}* were placed on each mouse, and the mutant-infected nymphs were allowed to feed on two different cohorts of mice (with or without IPTG). Pools of nymphs from each mouse were evaluated by semi-solid plating and qPCR for burdens (Figs. 3e, f and S5a-b; FN). Nymphs infected with *bb0401^{cond}* and fed on mice without IPTG contained DNA burdens equivalent to that of the pre-fed nymphs but failed to show proliferation of spirochetes similar to *wt*; plating also showed significantly lower viable burdens compared to DNA burdens demonstrating not just the inability to proliferate during feeding, but a significant loss of viable spirochetes. Alternatively, only trace amounts of DNA were detected for *opp^{cond+28-1}* fed on mice without IPTG and no viable colonies were found by plating. *Wt* and mutant-infected nymphs fed on mice supplemented with IPTG contained equivalent burdens by plating and qPCR. Again, colonies were recovered from plating, and all isolates were shown to be IPTG responsive confirming there were not spontaneous mutations. To evaluate transmission, we collected sera for western blotting (Fig. 3g, h) and tissues for culture (Table 1) and qPCR (Fig. 3i, j) at 2 wk post-infestation. We found that all mice infested with ticks colonized by *bb0401^{cond}* or *opp^{cond+28-1}* without IPTG-supplemented water displayed no seroreactivity, and tissue cultures were all negative at 2 wk post-inoculation. Mice that received IPTG-water had positive tissue cultures for both *bb0401^{cond}* and *opp^{cond+28-1}*, and tissues at 2 wk post-inoculation demonstrated qPCR burdens consistent with *wt* for all tissues, with only a small reduction in *opp^{cond+28-1}* skin and heart. Given the use of IPTG induction during the tick feeding cycle has not been utilized to our knowledge, we sought to detect IPTG in mouse plasma and ticks (Fig. S5d-e). IPTG was only able to be detected above the limit of quantification in mouse plasma and fed nymphs, though samples showed consistent amounts of IPTG within these groups. While we can extrapolate from the mouse plasma IPTG levels that larvae would have obtained some inducer during the bloodmeal, we cannot confirm whether residual IPTG was present in the molting nymph. Overall, these data demonstrate that both *bb0401* and *opp* are essential for replication within the tick and transmission to the host and confirms that the inducible system can be utilized throughout the enzootic cycle.

## BB0401 is a GltP

To understand how the loss of these amino acid transporters impacts the spirochete metabolically, we assayed a broad panel of metabolites for +/- IPTG cultures of *bb0401^{cond}* and *opp^{cond}* after 48 h of starvation (Figs. 4a, b, S6b-c, Supplementary data 1). As a control for any background metabolic shifts that arise from the use of the inducible system in *B. burgdorferi*, we included analysis of an inducible GFP (*igfp*) strain (Fig. S6a, Supplementary

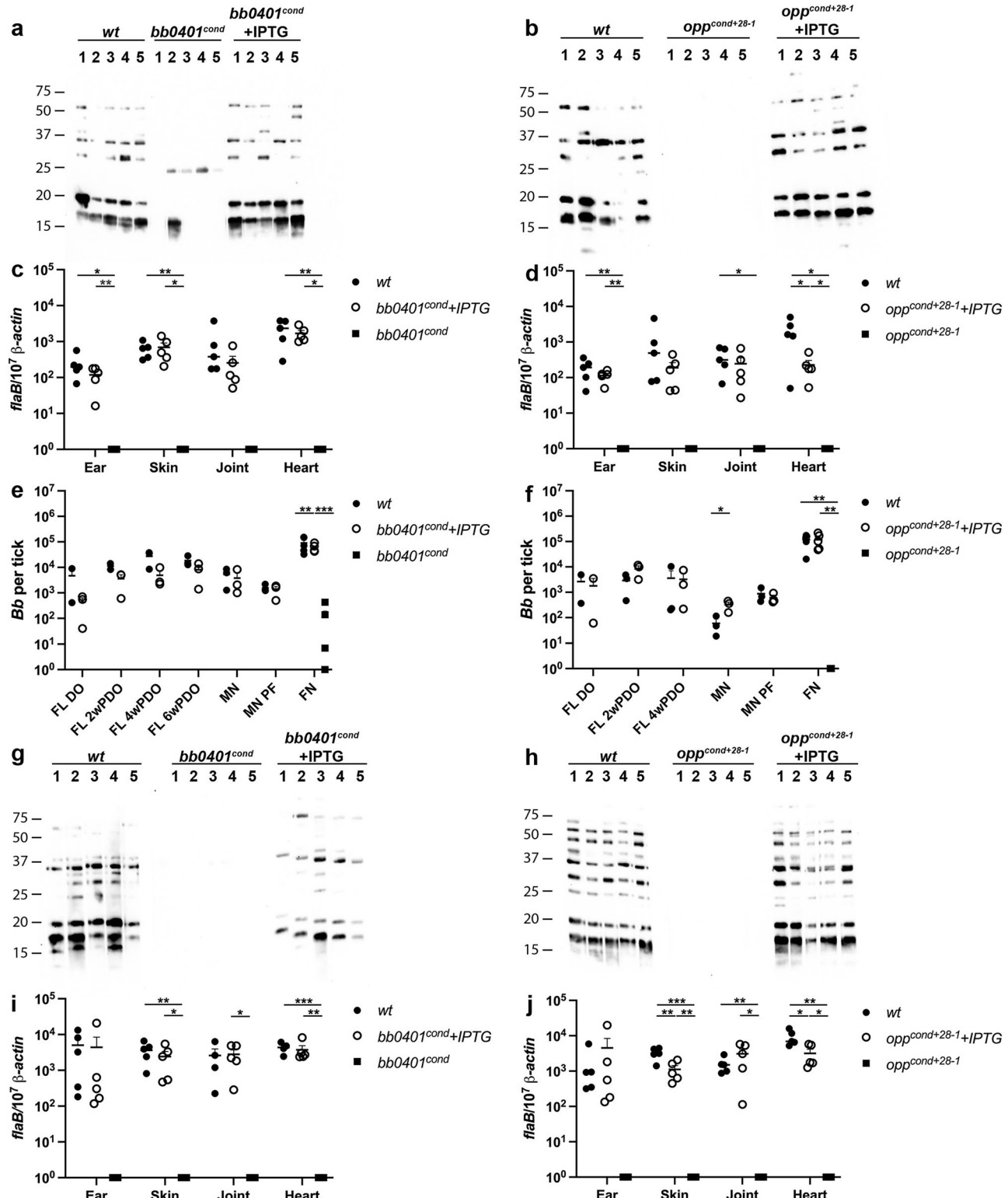

**Fig. 3 | *bb0401* and *opp* transporters are required for infection and transmission.** Western blots of sera against spirochete whole cell lysates for mice needle-inoculated with (**a**) *bb0401^cond* and (**b**) *opp^cond*. DNA burdens of tissues ($n = 5$) for mice 2 wk post-inoculation as determined by qPCR for (**c**) *bb0401^cond* and (**d**) *opp^cond*. Viability plating of fed larvae at drop-off (FL DO, $n = 3$), fed larvae at timepoints post-drop-off (FL #wPDO, $n = 3$), post-molt nymphs (MN, $n = 3$), molted nymphs prior to feeding (MN PF, $n = 3$), and fed nymphs (FN, $n = 5$) for (**e**) *bb0401^cond* and (**f**) *opp^cond*.

Western blots of sera against spirochete whole cell lysates for mice nymph-inoculated with (**g**) *bb0401^cond* and (**h**) *opp^cond*. DNA burdens of tissues ($n = 5$) for mice 2 wk post-infestation as determined by qPCR for (**i**) *bb0401^cond* and (**j**) *opp^cond*. Mouse cohorts that received IPTG are represented with a "+IPTG." All *p*-values were determined for pairwise comparisons using a two-tailed unpaired *t* test. *$p$ value <0.05, ** <0.01, *** <0.001, **** <0.0001. Exact *p* values can be found in Supplementary data 3.

**Table 1 | Tissue Culture for in vivo studies**

| 2wk post-inoculation | | | | | | |
|---|---|---|---|---|---|---|
| | *wt* | *bb0401^cond* | *bb0401^cond* + IPTG | *wt* | *opp^cond+lp28-1* | *opp^cond+lp28-1* + IPTG |
| Ear | 5/5 | 0/5 | 5/5 | 5/5 | 0/5 | 5/5 |
| Inoculation site | 5/5 | 0/5 | 5/5 | 3/5 | 0/5 | 5/5 |
| Tibiotarsal joint | 5/5 | 0/5 | 5/5 | 5/5 | 0/5 | 5/5 |
| Bladder | 5/5 | 0/5 | 5/5 | 4/5 | 0/5 | 5/5 |
| Heart | 5/5 | 0/5 | 5/5 | 4/5 | 0/5 | 5/5 |
| Total tissues | 25/25 | 0/25 | 25/25 | 21/25 | 0/25 | 25/25 |
| Total mice | 5/5 | 0/5 | 5/5 | 5/5 | 0/5 | 5/5 |
| **4wk post-inoculation** | | | | | | |
| | *wt* | *bb0401^cond* | *bb0401^cond* + IPTG | *wt* | *opp^cond+lp28-1* | *opp^cond+lp28-1* + IPTG |
| Ear | 4/4 | ND | 4/4 | 2/2 | ND | 3/3 |
| Inoculation site | 4/4 | ND | 4/4 | 2/2 | ND | 3/3 |
| Tibiotarsal joint | 4/4 | ND | 4/4 | 2/2 | ND | 3/3 |
| Bladder | 4/4 | ND | 4/4 | 2/2 | ND | 3/3 |
| Heart | 4/4 | ND | 4/4 | 2/2 | ND | 3/3 |
| Total tissues | 20/20 | ND | 20/20 | 10/10 | ND | 15/15 |
| Total mice | 4/4 | ND | 4/4 | 2/2 | ND | 3/3 |
| **2wk post-infestation** | | | | | | |
| | *wt* | *bb0401^cond* | *bb0401^cond* + IPTG | *wt* | *opp^cond+lp28-1* | *opp^cond+lp28-1* + IPTG |
| Ear | 5/5 | 0/5 | 5/5 | 5/5 | 0/5 | 4/5 |
| Inoculation site | 5/5 | 0/5 | 5/5 | 5/5 | 0/5 | 0/5 |
| Tibiotarsal joint | 5/5 | 0/5 | 5/5 | 5/5 | 0/5 | 5/5 |
| Bladder | 5/5 | 0/5 | 5/5 | 5/5 | 0/5 | 5/5 |
| Heart | 5/5 | 0/5 | 5/5 | 5/5 | 0/5 | 5/5 |
| Total tissues | 25/25 | 0/25 | 25/25 | 25/25 | 0/25 | 19/25 |
| Total mice | 5/5 | 0/5 | 5/5 | 5/5 | 0/5 | 5/5 |

data 1). The only metabolite that resulted in a value above the 10% false discovery rate cutoff (FDR; -log(p)=2.7) was citrulline (logFC=0.99) which was higher in the uninduced condition. Intracellular concentrations of 38 and 58 metabolites changed above the 10% FDR threshold for *bb0401^cond* and *opp^cond*, respectively (Fig. S6b-c, Supplementary data 1). Of these affected metabolites, 31 showed overlap between the two strains but only 27 of these overlapping metabolites were shifted in the same direction; notably, citrulline was higher in the uninduced sample as seen in the *igfp* control. Acetyl-CoA was increased during starvation of both mutants with an 8.0-fold increase in *bb0401^cond* and a 9.7-fold increase in *opp^cond*. Nucleotides were affected in both mutants in a consistent manner that reflected large scale energy depletion with nucleotide monophosphates increasing and nucleotide triphosphates decreasing under starvation conditions (Fig. S6b-c, Supplementary data 1). In addition to the conserved energy-associated pattern, strain-specific patterns in specialized nucleotide derivatives, including an increase in c-di-GMP (cdGMP), were observed in *opp^cond* but not *bb0401^cond* (Supplementary data 1). Glycolytic and pentose phosphate pathway (PPP) intermediates were largely reduced in both strains with a much stronger phenotype in these pathways for *opp^cond*. An exception was observed for fructose 1,6-bisphosphate (FBP) and dihydroxyacetone phosphate (DHAP) which were elevated specifically in *bb0401^cond* starvation. Of the amino acids, only aspartate (logFC = −0.33) was reduced in *bb0401^cond* in the starved condition, while a number of other amino acids were found to be more abundant (e.g. tryptophan). In contrast, *opp^cond* showed reduced levels of multiple amino acids in the starved state (tryptophan, tyrosine, phenylalanine, arginine, glutamine, isoleucine, ornithine, valine, and leucine; logFC between -0.35 and -2.11); only glutamate and serine were more abundant.

While the metabolomics data suggested that BB0401 may transport aspartate, compensation by the peptide transport system, as suggested by the contrasting pattern of tryptophan between the *bb0401^cond* and *opp^cond* mutants, complicates the direct metabolic readout using unlabeled substrates. Therefore, we sought to directly test the transport of glutamate and aspartate by BB0401. Using a heavy isotope labeling assay, we evaluated transport of the two amino acids in *bb0401^cond* under induced and uninduced conditions. To confirm we could detect and differentiate light and heavy amino acids, we tested uptake in *wt* after incubation with heavy or light amino acids for 2 h and were able to discriminate and detect transport of heavy amino acids (Fig. 4c, e). When we tested transport of these amino acids on *bb0401^cond* with IPTG or 48 h after removal of inducer (Fig. 4c, e), there was a small but statistically significant decrease in the intracellular abundance of these amino acids under the starved condition (1.4- and 1.1-fold decrease for aspartate and glutamate, respectively). Given *bb0401^cond* starvation timepoints were observed to only impact recovery of the bacteria starting at 7 d without inducer (Fig. 2c, d), we repeated the assay by testing transport of amino acids after 7 d of starvation, and we increased the labeling time to 6 h to prolong the time for amino acid flux (Fig. 4d, f). Under these conditions, we saw an increase in levels of heavy labeled amino acids as a percentage of the total amino acid signal, due to the longer exposure time, as well as a more dramatic reduction in the amount of heavy labeled aspartate and glutamate imported in the starved culture (2.5- and 2.3-fold for aspartate and glutamate, respectively). Heavy glutamate was found to be imported at significantly lower percentages of the total glutamate signal than heavy aspartate in all samples, consistent with reports for Glt_Ph and Glt_Tk preferences for aspartate[26,27,29].

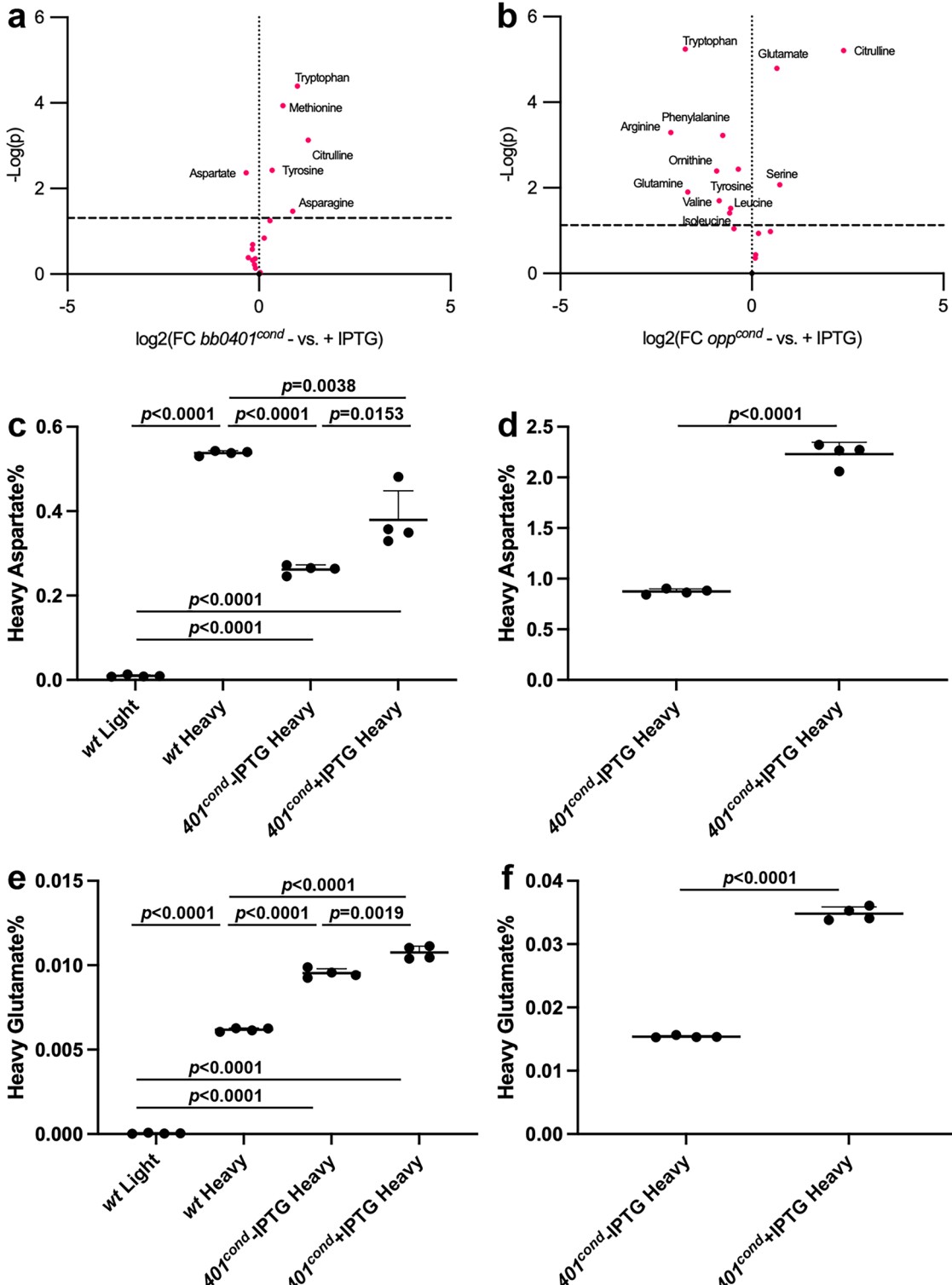

**Fig. 4 | *bb0401* is a GltP.** Volcano plots showing amino acid changes in (**a**) *bb0401cond*, and (**b**) *oppcond* when grown without and with 1 mM IPTG. Dotted line represents 10% FDR, *n* = 4. Tabulated results can be found in Supplementary data 1. Heavy amino acid transport assays for aspartate after (**c**) 48 h starvation and (**d**) 7 d starvation and glutamate after (**e**) 48 h starvation and (**f**) 7 d starvation of *bb0401cond*. *p*-values were determined for pairwise comparisons using a two-tailed unpaired *t* test, *n* = 4.

## Amino acid starvation does not stimulate a stringent response

*E. coli* utilizes the production of (p)ppGpp to initiate a stringent response under amino acid starvation conditions[10,47]. *B. burgdorferi* can synthesize and degrade (p)ppGpp using the homolog Rel_Bbu, which is required for the response to nutrient starvation in vitro, for survival in ticks between blood meals, and has been shown to modulate the spirochete's transcriptome[9,48]. To determine whether this response circuit for amino acid starvation was intact in *B. burgdorferi* we evaluated the transcriptome using RNAseq. When we first looked at RNA quality from *oppcond* cultured without inducer for either two or five days, we found that prolonged starvation resulted in

significant RNA degradation (Fig. S7). RNA degradation also appeared to be problematic in quantification of *bb0401* transcript during starvation, though it is unclear whether this phenotype is unique to amino acid starvation models. Given these results we chose to incubate both mutants without IPTG for two days to standardize the comparison between the two strains and to utilize a timepoint at which both mutants began to show effects of starvation but minimally impacted RNA quality. RNAseq was performed on cultures with and without IPTG for each strain in quadruplicate (Supplementary data 2). Transcriptional changes for *bb0401^cond* +/- IPTG were unremarkable, with *bb0401* being the only gene with a |logFC| >1 and a $p_{adj}$ < 0.05 (logFC = −3.325). *opp^cond* +/- IPTG demonstrated a larger transcriptional response of 102 genes with a |logFC| >1 and $p_{adj}$ < 0.05; 53 genes were up-regulated and 49 were down-regulated under starvation with the greatest fold change for BBD24 (logFC=3.08). The targets of the *opp^cond* mutagenesis, *oppD* and *oppF* (logFC = −2.59 and −1.42, respectively), were down-regulated during starvation. Interestingly, some of the substrate binding proteins for the *opp* system were differentially regulated with *oppA1* and *oppA3* down-regulated (logFC = -1.55 and -1.22, respectively) and *oppA4* was up-regulated (logFC=1.17).

Given that *B. burgdorferi* encodes three major regulatory systems—Histidine kinase-1/Response regulatory protein-1 (Hk1/Rrp1; BB0420/BB0419) and its product c-di-GMP, RpoN/RpoS, Rel_Bbu and its product (p)ppGpp—we compared the *opp^cond* transcriptome to known regulons to determine which regulatory schemes may be amino acid responsive. The only significant overlap was found with the Rel_Bbu regulon as defined by Drecktrah et al.[9]. The Rel_Bbu regulon was evaluated under several conditions (stationary growth, media starvation, and starvation recovery) with transcriptional variations for each condition; therefore, to simplify the comparison, we tabulated the Rel_Bbu regulon as any gene under the three conditions that demonstrated a |logFC| >1. Using these criteria, the Rel_Bbu regulon consists of 511 genes with only 42 genes overlapping differential expression in *opp^cond*. Of these overlapping genes, only 20 genes were regulated in the same direction due to loss of (p)ppGpp synthesis or amino acid starvation while the remainder were inversely regulated between the two conditions. DksA is also known to work cooperatively with (p)ppGpp or independently to facilitate interactions with RNAP for transcriptional control[12]. Of the 454 genes that define the DksA regulon[13], only 43 genes overlap with the differentially regulated genes for *opp^cond* and only 3 genes were regulated in the same direction due to loss of DksA or amino acid starvation. Twenty-one genes in this overlap were unique to the DksA regulon and not found in Rel_Bbu.

The inability to transcriptionally respond to severe amino acid starvation strongly suggests that the (p)ppGpp response in *B. burgdorferi* is uncoupled from amino acid sensing mechanisms. However, with the small amount of transcriptional overlap, we sought to quantify the production of (p)ppGpp during amino acid starvation using ³²P-labeling and thin layer chromatography (Fig. 5). We first confirmed that IPTG treatment of *wt* cultures did not result in changes to (p)ppGpp levels. *opp^cond* was grown under limiting starvation conditions (0.08 mM IPTG, 3 d) and severe starvation conditions (0 mM, 6 d and 7 d)[19] to evaluate the (p)ppGpp response. Consistent with our transcriptomic data, (p)ppGpp levels did not increase during starvation, instead they appeared to decrease under limiting starvation and could no longer be resolved under severe starvation, suggesting the stringent response was not activated under these conditions.

## Discussion

As an extreme auxotroph, *B. burgdorferi* likely contains many transport systems that are critical for survival[3]. Investigation of these systems are often stymied by limitations in manipulating the complex, partially defined in vitro growth medium and system redundancies encoded in the genome. Herein, we have characterized the contributions of a multipartite oligopeptide transport system and an aspartate/glutamate transporter to meeting the spirochete's amino acid needs. We previously demonstrated that *B. burgdorferi* relies on peptide transport for viability[19], a seemingly expedient approach to amino acid acquisition in the absence of de novo synthesis

pathways. However, we hypothesized that, of the handful of free amino acid transporters encoded in the *B. burgdorferi* genome, we would likely find other essential amino acid acquisition pathways, either to complement amino acid biases related to available peptide composition or for amino acids required for non-proteinogenic roles in the spirochete's physiology.

There has been a growing list of genes required for *B. burgdorferi* viability that include the response regulator Rrp2, BamA porin, DedA, telomere resolvase ResT, FtsH protease, RNaseY, Ldh, and BBD18[37–44]. Herein, we find that both Opp and BB0401 are critical transporters for the spirochete both in vitro and in vivo. While GltPs have been shown to provide ancillary amino acid support for bacteria[49–51], *B. burgdorferi* is the only bacteria to date that has been shown to require it for viability, similar to the Opp system. Superficially, both mutants display growth and morphology phenotypes that appear similar. It is likely that the subtle differences in IPTG concentrations required for normal growth are reflective of small differences in expression levels needed to meet the spirochetes' acquisition needs. However, the ability of *bb0401^cond* to survive starvation conditions for significantly longer than *opp^cond* could be indicative of a technical limitation such as RNA or protein stability or the severity of the metabolic lesion such as narrow versus broad amino acid starvations. Interestingly, *opp^cond* spirochetes retain motility during starvation, though the motility is highly dysregulated due to aberrant morphologies[19]. In contrast, *bb0401^cond* quickly became non-motile despite maintenance of the cell's planar waveform. Indeed, the "elongation" noted for *bb0401^cond* is likely a result of this lack of motility. As seen with the *flaB* mutant[52], without the ability to swim in opposing direction during replication, *B. burgdorferi* cells will often grow in chains where septa can be distinguished at regular intervals as seen with *bb0401^cond*. If these chains represent individual cell division events as expected, then our growth curves are likely underestimating *bb0401^cond* growth potential, as long spirochetes were enumerated as single cells. Spirochete morphology and motility is dependent on their endoplasmic flagella and peptidoglycan layer[52], the latter of which requires glutamate. While we have shown that BB0401 transports both glutamate, one would expect significant perturbation of the cell's peptidoglycan would result in other morphological anomalies and the lack thereof may suggest a more thorough evaluation of the peptidoglycan composition may be warranted.

Our study of these transporters in vivo required the use of a supplemental IPTG infection model. We were able to show that *ad libitum* supplementation of IPTG via water was sufficient to complement expression of both *bb0401* and *opp* in the conditional mutants in all tissues collected up to 4 wks during murine infection. We were also able to demonstrate IPTG-induction of *bb0401* in vivo resulted in expression levels similar to *wt*. We were unsure whether these mutants would survive within the tick through the molting phase, as we had no way to supplement IPTG outside of the ticks' bloodmeal; however, we found that both mutants survived all stages of digestion and molting. While we could not rule out residual IPTG (or other lactose analogs) in the tick due to a high threshold of detection, our room temperature culture experiment (Fig. S4c) strongly suggests that these systems are not critical in the absence of proliferation stimuli. Therefore, the two mutants likely survived within the tick for ~2-2.5 months post-repletion due to bacterial quiescence in the molting nymph and these data shed more insight on the metabolic state of the bacteria in these understudied conditions. Regardless, upon taking the nymphal bloodmeal, these mutants were unable to transition to peak replication phases and resulted in little to no viable bacteria post-feeding. Notably, not only did the mutant spirochetes fail to proliferate during nymphal feeding, but they also failed to survive within the midgut; *opp^cond* tick plating resulted in no viable spirochetes and *bb0401^cond* showed fewer viable spirochetes than the unfed nymph. While there were some viable spirochetes in *bb0401^cond* nymphs, it is likely that the motility defect would have prevented migration out if the midgut and into the skin. Notably, IPTG-supplementation was sufficient to complement both strains in the tick and during transmission and infection. These data

demonstrate that IPTG-supplementation during the enzootic cycle is a viable model for evaluating conditional mutants at each stage of the cycle.

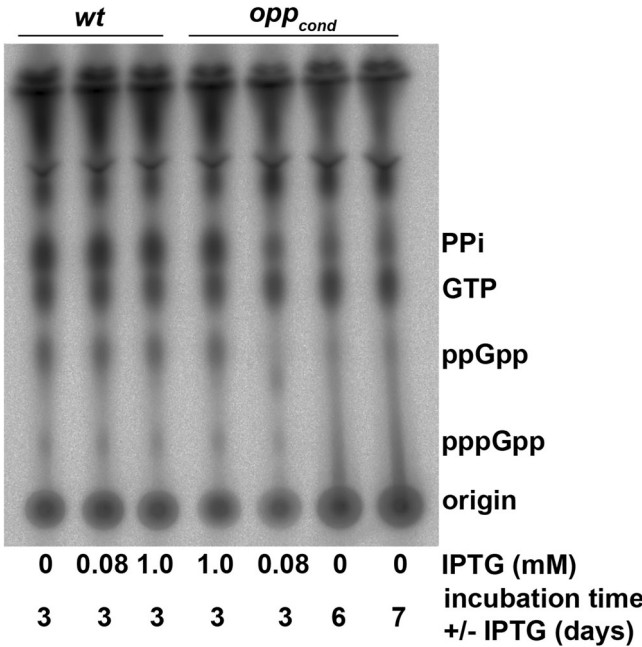

**Fig. 5 | Abrogation of peptide transport does not result in a (p)ppGpp response.** Thin layer chromatography analysis of $^{32}P$-orthophosphate labeled *opp$^{cond}$* cultures with or without IPTG. *wt* controls with and without IPTG were included to establish baseline (p)ppGpp production.

We were able to experimentally confirm that BB0401 functions as a GltP. Given the detrimental phenotype of *bb0401$^{cond}$* and the homology of BB0729 to TcyP transporters, we believe BB0401 to be the only GltP homolog in *B. burgdorferi* and propose designating BB0401 as GltP$_{Bb}$; Fig. 6 summarizes our current knowledge of amino acid transport in the spirochete. Of note, GltP$_{Bb}$ preferentially transported aspartate though it appears to retain low affinity for glutamate, which is in contrast to the other characterized homologs which either exclusively transport aspartate (Glt$_{Ph}$ and Glt$_{Tk}$) or display equivalent transport of both glutamate and aspartate (GltP$_{Ec}$) and may contribute to the overall lack of binding residue conservation or the predicted consolidation of the binding site for both amino acids in contrast to GltP$_{Ec}$. Metabolomics also demonstrated the extent of amino acid starvation for *opp$^{cond}$*, allowing us to extrapolate which amino acids are not being transported via the remaining free amino acid transporters encoded by *B. burgdorferi*. BB0401 is one of the few transporters that have highly conserved substrates; therefore, these data will be helpful to screen for function related to the remaining transporters. While we saw other metabolic shifts due to these targeted starvations, the metabolic perturbations were surprisingly narrow in both shared and unique changes between the two starvation types. Interestingly, norepinephrine, which is synthesized from tyrosine, was lower in starved *opp$^{cond}$* as was tyrosine, suggesting that *opp* could have some role in transporting amino acids derivatives, though our limited metabolomics approach would likely have missed a number of these targets. Norepinephrine had previously been demonstrated to play a role in up-regulation of OspA[53], a tick phase surface protein, and has been implicated in virulence of other pathogens[54]. It is unclear which component of BSK-II would provide norepinephrine during in vitro culture, though the medium does contain rabbit serum. It is also possible that we detected a norepinephrine analog that could not be resolved using this metabolomics approach and warrants further studies. Nucleoside mono-, di-, and triphosphates were broadly affected in both strains, with many triphosphates lower under starvation conditions and many

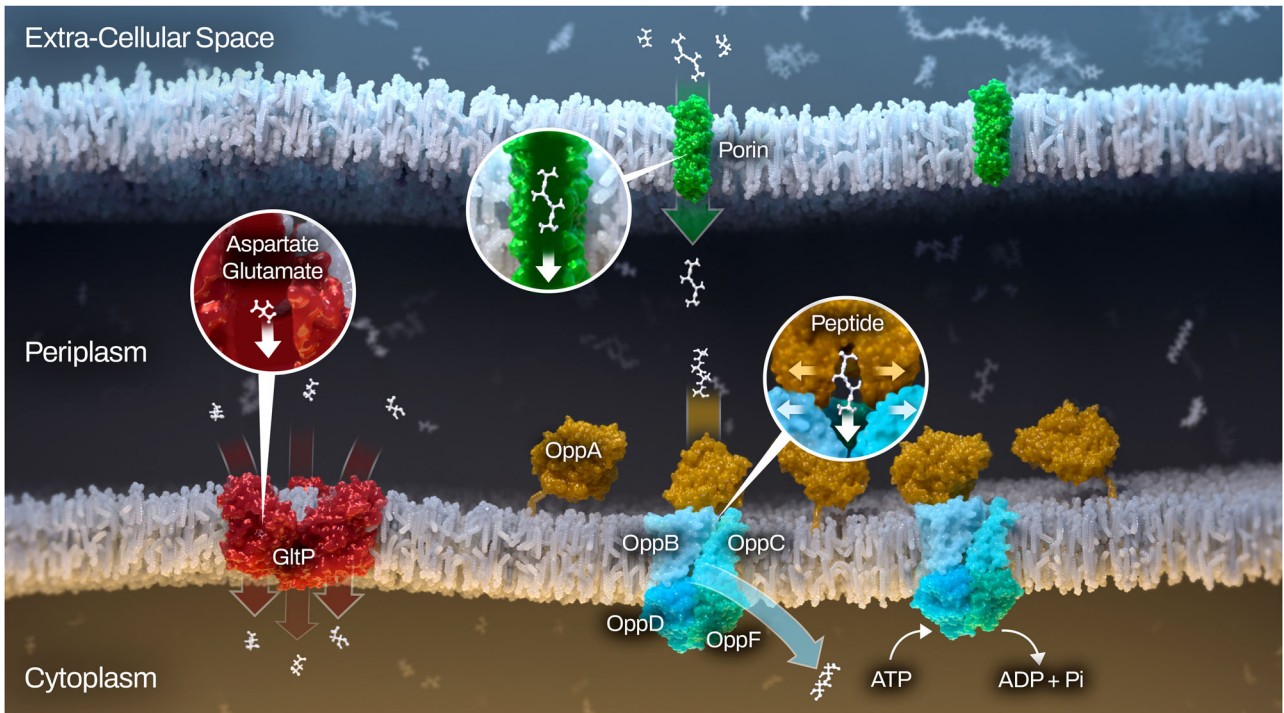

**Fig. 6 | Graphical summary of confirmed amino acid transport mechanisms for *B. burgdorferi*.** Non-specific porins (green) in the outer membrane facilitate peptide and amino acid flux into the periplasm and may provide a limitation in peptide size during passive diffusion. The five OppAs (OppA1-5; orange) are lipoproteins tethered to the inner membrane where upon binding peptides in the periplasm they dock to the Opp permease (OppBC; light blue) and transport their cargo to the cytoplasm. The change in permease conformation is facilitated by hydrolysis of ATP by the nucleotide binding domain (OppDF; blue and green). Alternatively, GltP (red) can symport free glutamate or aspartate along with protons and/or sodium ions.

mono-/diphosphates higher, indicating a loss of energetic potential. Of note, *opp* is an ATP-driven transporter where GltP$_{Bb}$ is a symporter, but ATP was lower with starvation of both mutants, suggesting that ablation of NBD-binding in *opp$^{cond}$* was not generally responsible for these shifts. Potentially contributing to the drop in nucleoside triphosphates is perturbation of the glycolysis pathway in both strains, which serves as the primary energy generation pathway for *B. burgdorferi*[3].

As had been inferred from the significant maladaptive responses in the *opp$^{cond}$* mutant[19], we found that there is no significant stringent response initiated in *B. burgdorferi* upon amino acid starvation, despite having the machinery to generate and utilize the alarmone (p)ppGpp. While the lack of de novo amino acid synthesis would prevent the spirochete from up-regulating enzyme pathways to supplement amino acids via synthesis, control over aspects such as cell growth is also lost. As a result, we find *opp$^{cond}$* suffers from uncontrolled cell elongation under starvation conditions and, though cells may retain viability for a short period of time, these cells lose the ability to recover from starvation quickly. The inability to mitigate amino acid injury by the bacteria is likely unimportant in the context of the enzootic cycle, as the predictable nutritional niches occupied by the spirochete are not specifically deficient in amino acids; therefore, one of the other limited nutrients likely initiates the stringent response. However, in the context of bacterial inhibitors, the failure to respond to amino acid stress presents a novel opportunity to develop targeted and unique approaches to bacterial inhibition. Following this logic, we published a proof-of-concept study that showed the *opp* system could be targeted by small molecules that resulted in bacterial growth inhibition[55]. The GltP$_{Bb}$ could be targeted in concert with Opp to increase bacterial inhibition. While EAATs are critical to the host[24] and could complicate off-target effects of inhibiting GltP$_{Bb}$, the lack of conservation among predicted binding pocket residues may help discriminate between the two. The development of more targeted treatment approaches could shift the clinical landscape for Lyme disease treatment, specifically in relation to Lyme-related complications such as persistent infection or post-treatment Lyme disease.

## Materials and methods
### Sequence alignments and phylogeny
Multiple sequence alignments of proteins were generated using Multiple Sequence Comparison by Log-Expectation (MUSCLE)[56]. The output file was submitted to Phylogenetic Maximum Likelihood (PhyML)[57] for phylogenetic analysis and the output phylogenetic tree was displayed using the Interactive Tree of Life (iTOL)[58]. The list of proteins and their UniProt IDs can be found in Table S3.

### Modeling and docking
BB0401 and GltP$_{Ec}$ was modeled after the Glt$_{Ph}$ crystal structure (PDB ID 2nwl)[26] using SwissModel[32] or de novo using AlphaFold2-Multimer[35]. ChimeraX[59] was used to calculate Root Mean Square Deviation (RMSD) values. SwissDock[60] was used to dock glutamate and aspartate in the homology model of BB0401. Structures were visualized in either PyMol[61] or ChimeraX. The crystal structure of Glt$_{Ph}$ bound to sodium ions (PDB ID 2nwx)[26] was used for the overlay imaging examining conservation of sodium binding sites. The GltP$_{Ec}$ binding residues were identified in a previous study by Rahman et al.[24].

### Bacterial strains and culture conditions
*Escherichia coli* strains Top10 and Stellar (Table S1) were grown in Luria-Bertani (LB) broth or on LB plates with appropriate antibiotics (ampicillin [Amp; 100 μg/ml], spectinomycin [Spec; 100 μg/ml], or gentamicin [Gent; 5 μg/ml]) at 37 °C. All strains of *Borrelia burgdorferi* used in this study are derived from B31 5A18 NP1[62] (Table S1), which has a kanamycin selection cassette inserted in lp25 and is missing lp56 and lp28-4 (Fig. S3). *B. burgdorferi* strains were cultured in modified Barbour-Stoenner-Kelly II (BSK-II) medium[63] supplemented with 6% rabbit serum and relevant antibiotics (kanamycin [Kan; 400 μg/ml], streptomycin [Strep; 50 μg/ml],

erythromycin [Erm; 0.06 μg/ml] or gentamicin [Gent; 50 μg/ml]) at 37 °C in a $CO_2$ incubator (5%) unless otherwise noted. Conditional lethal mutants were grown with the addition of 1 mM isopropyl-β-d-thiogalactoside (IPTG; GoldBio) unless otherwise noted. Conditional lethal mutants without IPTG were maintained in minimal volumes and low densities to prevent spontaneous mutations in the *lac* operator sequence and the outgrowth of constitutively active transcription of target genes as has been previously seen in conditional lethal *Borrelia* mutants[41].

### Generation of plasmids and mutant strains
*B. burgdorferi* B31 chromosomal sequence AE000873.1 was used as reference for all cloning. All plasmids and strains used in this study are referenced in Table S1. All oligonucleotides used in this study are referenced in Table S2. **pibb0401 Shuttle Vector**. The ORF of *bb0401* was amplified using 5'/3' ibb0401 (1232 bp), which included 15 bp overlap with the 5' and 3' restriction sites of the targeted cloning vector. The IPTG-inducible, cp9-based shuttle vector containing a Spec/Strep selection cassette (P$_{flgB}$-*aadA*), pJSB275[39], was digested with NdeI/HindIII to release the luciferase ORF and linearize the vector backbone. The *bb0401* ORF and pJSB275 vector backbone were joined using InFusion EcoDry (Clonetech), transformed into chemically competent Stellar *E. coli* (Clonetech), and positive clones were selected with Spec. A positive clone was identified by PCR using 5'/3' pJSB275 seq and sequence was confirmed by Sanger sequencing. The resulting plasmid was designated pi*bb0401* (pEcAG286). **pΔbb0401 Suicide Vector**. The regions upstream (F1; 1073 bp) and downstream (F2; 1073 bp) of *bb401* were amplified using 5'/3' bb0401 null F1 and 5'/3' bb0401 null F2, respectively. The P$_{flgB}$-*aacC1* Gent cassette (966 bp) was amplified from pBRV2 using 5'/3' bb0401 null gent. Primers for all three fragments included 15 bp overlaps with adjacent fragments and were assembled into BamHI-digested pUC19 using InFusion EcoDry, transformed into chemically competent Stellar *E. coli*, and positive clones were selected with Gent. A positive clone was identified by PCR using pless Gent F/R and sequence was confirmed by Sanger sequencing. The resulting plasmid was designated pΔ*bb0401* (pEcAG259) **ibb0401 Mutant**. Competent *B. burgdorferi* B31 5A18 NP1[62] was electroporated with pi*bb0401* as previously described[64] and transformants were selected in BSK-II with Kan and Strep. Clones were screened for the presence of *aadA* with pless Strep F/R and DNA from transformants were subsequently transformed into *E. coli* Top10 to confirm recovery of shuttle vector. Selected clones were screened for endogenous plasmid content using a modification of the plasmid multiplex PCR described in Bunikis et al.[65,66]. A single clone was selected for subsequent cloning and designated *ibb0401*. **bb0401$^{cond}$ Mutant**. The conditional *bb0401* mutant was generated by electroporating competent *ibb0401* with linearized pΔ*bb0401* and transformants were selected in BSK-II with Kan, Strep, Gent, and 1 mM IPTG. Single clones were screened for the presence of *aacC1* with pless Gent F/R. Selected clones were screened for endogenous plasmid content using a modification of the plasmid multiplex PCR[65,66]. A clone with all parental plasmids except cp32-3, cp9, and lp5, plasmids known to be dispensable for the enzootic cycle[36], was selected for characterization and designated *bb0401$^{cond}$* (Fig. S3). **pigfp Shuttle Vector**. The ORF of *gfp* was amplified using 5'/3' iGFP (750 bp), which included 15 bp overlap with the 5' and 3' restriction sites of the targeted cloning vector and was cloned as described above for pi*bb0401*. A positive clone was identified by PCR using 5'/3' pJSB275 seq and sequence was confirmed by Sanger sequencing. The resulting plasmid was designated pi*bb0401* (pEcAG304). **igfp Strain**. Competent *B. burgdorferi* B31 5A4[36] was electroporated with pi*gfp* as previously described[64] and transformants were selected in BSK-II with Strep. Clones were screened for the presence of *aadA* with pless Strep F/R and DNA from transformants were subsequently transformed into *E. coli* Top10 to confirm recovery of shuttle vector. **qPCR standards**. 5'/3' primers for *flaB*, *bb0401*, and mouse β-actin were used to amplify target coding regions for each gene and fragments were cloned into pCR2.1 using the TOPO Cloning Kit (Thermo). Individual clones were confirmed via Sanger sequencing.

## Growth kinetics

$bb0401^{cond}$ growth was evaluated similar to that of $opp^{cond}$[19]. In brief, cultures were grown in BSK-II medium with 1 mM IPTG and appropriate antibiotics to mid-logarithmic growth. Cultures were washed with PBS to remove IPTG and resuspended in BSK-II at the density of $1 \times 10^6$ or $1 \times 10^3$ cells/ml respectively with varying IPTG concentrations (0–1 mM). In case of recovery analysis, cultures were washed as described above and resuspended in BSK-II media without IPTG and incubated at 37 °C; each day, 1 mM IPTG was added to aliquots of cultures to determine their ability to recover from lack of induction. All growth experiments were performed in triplicate and cell density was enumerated daily by dark-field microscopy. Growth curves were plotted using Prism software (v9.5.1; GraphPad Software, Inc.) and are representative of three individual experiments.

## Microscopy and measurement of spirochete length

$Wt$ and $bb0401^{cond}$ cell lengths were evaluated similar to that of $opp^{cond}$[19]. Strains were grown in BSK-II medium ($bb0401^{cond}$ with 1 mM IPTG) with appropriate antibiotics to mid-logarithmic growth. Cultures were washed with PBS and resuspended in BSK-II at the density of $1 \times 10^6$ spirochetes/ml in the absence of IPTG for $wt$ and $bb0401^{cond}$ or with 1 mM IPTG for $bb0401^{cond}$. Images were acquired daily using a 40x objective and a minimum of 100 spirochete lengths were measured for each sample using ImageJ 1.54 g. Images to inspect individual cells were collected using a 100x objective and processed using ImageJ 1.54 g. These data were graphed as whisker plots using Prism software. Images were collected with a Zeiss Axiocam 208 color camera (Zeiss).

## RT-PCR

For quantification of $bb0401$ transcript by qRT-PCR in vitro, $wt$ and $bb0401^{cond}$ strains were grown in BSK-II medium with 1 mM IPTG for the mutant strain and appropriate antibiotics to mid-logarithmic growth. Cultures were washed with PBS to remove IPTG and resuspended in BSK-II at the density of $1 \times 10^6$ spirochetes/ml in triplicate and growth at 37 °C for 2 d. $bb0401^{cond}$ was growth with 1 mM, 0.1 mM, 0.01 mM or without IPTG. Samples were collected and RNA was purified using the Zymo Direct-zol RNA miniprep kit (Zymo Research) according to manufacturer's protocols. Contaminating DNA was removed using TURBO DNA-free Kit (Thermo). Equivalent amounts of RNA were converted to cDNA using the SuperScript First-Strand Synthesis System (Thermo) and '3 primers for the $flaB$ and $bb0401$ ORFs (Table S2). Equivalent amounts of cDNA were subjected to PCR using 5'/3' primers for the $flab$ and $bb0401$ ORFs (Table S2) and Apex Red Master Mix (Genesee Scientific).

## Flow cytometry

$wt$, $opp^{cond}$, and $bb0401^{cond}$ were grown in BSK-II medium with 1 mM IPTG and appropriate antibiotics to mid-logarithmic growth. Cultures were washed with PBS to remove IPTG and resuspended in BSK-II at the density of $1 \times 10^4$ spirochetes/ml for $wt$ and $bb0401^{cond}$ and $1 \times 10^6$ spirochetes/ml for $opp^{cond}$ and incubated at 37 °C or $1 \times 10^7$ spirochetes/ml for all strains at room temperature incubation. The cell viability was tested using BDFacs Symphony cytometer using standard live/dead stains, Syto 9 (1 µM) (Thermo Fisher Scientific) propidium iodide (5 µM) (Thermo Fisher Scientific). Samples were stained and analyzed directly in BSK-II using the gating strategy described in Fig. S4a-b. Particulate matter was identified and excluded using forward- and side-scatter then bacterial cells were gated using Syto9 (FITC channel) and propidium iodide (TxRed-mCh channel).

## Ethics statement

All animal work was conducted according to the guidelines of the National Institutes of Health, *Public Health Service Policy on Humane Care and Use of Laboratory Animals*, and the United States Institute of Laboratory Animal Resources and National Research Council, *Guide for the Care and Use of Laboratory Animals*[67]. All protocols were approved by the Rocky Mountain Laboratories, NIAID, NIH Animal Care and Use Committee (2021-041E). The Rocky Mountain Laboratories are accredited by the International Association for Assessment and Accreditation of Laboratory Animal Care (AAALAC). We have complied with all relevant ethical regulations for animal use.

## Infection studies

All infections were performed on female C3H/HeJ mice aged five to eight weeks (Jackson Laboratories). Cohorts of 5 mice were utilized for 2wk infection timepoints for both needle- and nymph-inoculation based on a Fisher's Exact Test of determining infection (99%) or no-infection (1%) probability with an alpha of 0.01 and a power of 90. Our 4wk mouse infection cohorts were sized to generate enough ticks for subsequent analysis. No exclusion criteria, randomization, or blinding strategies were used during the experiment. *B. burgdorferi* strains were used to needle-inoculate cohorts of mice intradermally or infected nymphs ($n = 10$) were placed on mice as described below. For $opp^{cond}$ and $bb0401^{cond}$, cohorts of mice were either given IPTG water (2% sucrose, 80 mM IPTG) or water alone starting a week before infection, and this was maintained throughout the duration of the study. Tissues (ear, inoculation site skin, tibiotarsal joint, bladder, and heart) were collected for qPCR and cultured in BSK-II supplemented with BAM (*Borrelia* antibiotic mixture; 0.05 mg/ml sulfamethoxazole, 0.02 mg/ml phosphomycin, 0.05 mg/ml rifampicin, 0.01 mg/ml trimethoprim, and 2.5 µg/ml amphotericin B) and 1 mM IPTG (for mutants only). DNA was isolated from qPCR tissue samples using DNeasy Blood and Tissue Kit (Qiagen) modified by pre-digestion with collagenase IV (Sigma) and proteinase K (Sigma). qPCR assays were conducted using TaqMan Fast Advanced Master Mix (Thermo) on a QuantStudio 5 (Thermo). At two- and four-weeks post-infection mouse sera was collected and blotted at 1:1000 against *B. burgdorferi* whole cell lysates on nitrocellulose, followed by goat anti-mouse HRP at 1:15000 (Southern Biotech). Blots were developed with SuperSignal West Pico PLUS (Thermo) and imaged on iBright 1500 (Invitrogen).

## qRT-PCR

For quantification of $bb0401$ transcript by qRT-PCR in vivo, infected tissues were disrupted via the Qiagen TissueLyser LT and RNA was purified using the Zymo Direct-zol RNA miniprep kit (Zymo Research) according to manufacturer's protocols. Contaminating DNA was removed using TURBO DNA-free Kit (Thermo). Plus and minus RT cDNA libraries were made using SuperScript IV VILO Master Mix (Thermo) and qPCR was performed using TaqMan Fast Advanced Master Mix (Thermo) and primers/probes listed in Table S2 on a QuantStudio 5 (Thermo) with plasmids containing the target ORF serving as quantification standards.

## Tick studies

Pathogen-free *Ixodes scapularis* larvae (Oklahoma State University) were allowed to naturally acquire a bloodmeal from infected mice at two-weeks post-infection via whole body infestation. Fed larvae were allowed to molt into flat nymphs and were subsequently fed on naïve mice via capsule feeding[68] to measure transmission. Pooled ticks at all stages were evaluated for spirochete burdens via semi-solid plating[64] and qPCR[69] as previously described. Semi-solid plating media for all strains included BAM and conditional mutants were also grown with 1 mM IPTG. qPCR assays were conducted using TaqMan Fast Advanced Master Mix on a QuantStudio 5.

## RNAseq

To test RNA quality, $opp^{cond}$, the most impaired mutant, was grown BSK-II medium with 1 mM IPTG and appropriate antibiotics to mid-logarithmic growth. Cultures were washed with PBS to remove IPTG and resuspended in BSK-II at the density of $1 \times 10^7$ spirochetes/ml with or without 1 mM IPTG and grown at 37 °C for 2 d or 5 d in quadruplicate. Samples were analyzed on the BioRad Experion according to manufacturer's protocols. For RNAseq analysis $opp^{cond}$ and $bb0401^{cond}$ were grown as described but incubated for only 2 d in quadruplicate. Cell pellets were collected, and samples were flash frozen in liquid nitrogen and sent to SeqCenter for RNAseq analysis according to their standard prokaryotic protocols

and genes on plasmids that were not present in all samples were excluded from comparison. The full data table is available as Supplementary data 2.

## Metabolomics

*opp^cond* and *bb0401^cond* were grown in BSK-II medium with 1 mM IPTG and appropriate antibiotics to mid-logarithmic growth. Cultures were washed with PBS to remove IPTG and resuspended in BSK-II at the density of $1 \times 10^7$ spirochetes/ml with or without 1 mM IPTG and grown at 37 °C for 48 h in quadruplicate. Sample were then washed 0.9% sodium chloride solution (Teknova) and treated with LCMS grade methanol (Fisher Scientific), incubated on ice, and an equal volume of LCMS grade water (Fisher Scientific) was added. Samples were agitated to ensure layer mixing for 30 m under refrigeration and centrifuged at $16,000 \times g$ for 20 min. 400 μl of the top (aqueous) layer was collected for LCMS analysis of central polar metabolites.

Aqueous metabolites were analyzed via liquid chromatography tandem mass spectrometry (LC-MS/MS) using a combination of two analytical methods with opposing ionization polarities[70,71]. All methods used a LD40 XR UHPLC (Shimadzu Co.) system for separation and a 6500+ QTrap mass spectrometer (AB Sciex Pte. Ltd.) for detection and quantification. Negative mode samples were eluted from a Waters™ Atlantis T3 column (100 Å, 3 μm, 3 mm × 100 mm) using a binary gradient from 5 mM tributylamine, 5 mM acetic acid in 2% isopropanol, 5% methanol, 93% water (v/v) to 100% isopropanol over 5 m. Two distinct multiple reaction monitoring (MRM) pairs in negative mode were used for each metabolite. Positive mode samples were eluted from a Phenomenex Kinetex F5 column (100 Å, 2.6 μm, 100 × 2.1 mm) with a gradient from 0.1% formic acid in water to 0.1% formic acid in acetonitrile over 5 m. Tributylamine and all synthetic molecular references were purchased from Millipore Sigma. LCMS grade water, methanol, isopropanol and acetic acid were purchased through Fisher Scientific.

All signals were integrated using SciexOS 3.1 (AB Sciex Pte. Ltd.). Signals were confirmed visually. Signals with greater than 50% missing values were discarded and remaining missing values were replaced with the lowest registered signal value. Filtered datasets of the negative mode aqueous metabolites total sum normalized after initial filtering. The positive mode aqueous metabolomics dataset was scaled and combined with the negative mode aqueous metabolite dataset using common signals for tyrosine and phenylalanine to generate scalars between the datasets. A Benjamini-Hochberg method for correction for multiple comparisons was imposed to establish false discovery rate (FDR) cut-offs for significance.

## Amino acid uptake assay

*opp^cond* and *bb0401^cond* were grown in BSK-II medium with 1 mM IPTG and appropriate antibiotics to mid-logarithmic growth. Cultures were washed with PBS to remove IPTG and resuspended in BSK-II at the density of $1 \times 10^7$ spirochetes/ml with or without 1 mM IPTG and grown at 37 °C for 48 h or 7 d in quadruplicate. An equal density of *wt* cells were prepared for light and heavy controls. To compete with unlabeled amino acids, we added 1:1 heavy amino acids as defined by the CMRL content. Therefore, when cultures were ready, 30 mg/l of heavy labelled aspartic acid and 86.2 mg/l of heavy labelled glutamic acid and cultures were incubated for 2 h or 6 h as noted. Samples were then prepared and extracted as detailed above.

Novel MRMs were developed via direct injection of authentic standards. Heavy glutamate ($^{13}C_5,^{15}N_1$) and heavy aspartate ($^{13}C_4,^{15}N_1$) were purchased from Cambridge Isotope Laboratories, Inc. Homologous signals were confirmed for heavy isotopes to maintain comparisons to normal isotope metabolites. All metabolites were detected in negative mode according to the same LC-MS/MS parameters detains above for negative mode metabolomics profiling. Aspartate was detected via 132- > 88 and 137- > 92 for normal and heavy isotope respectively. Glutamate was detected via 146- > 102 and 152- > 107 for normal and heavy isotope respectively. Glucose was detected via 179- > 89 and 185- > 92 for normal and heavy isotope respectively.

## IPTG detection assay

Samples of 20 μl pooled sera (1 wk after starting IPTG-supplementation), 5 fed larvae, 10 unfed nymphs, or 3 fed nymphs were prepared and extracted as detailed above. IPTG standard was purchased from Gold Bio and detected in negative mode according to the LC-MS/MS parameters described above for negative mode metabolomics profiling. IPTG was detected with dual MRMs (237- > 161, 237- > 59) and signal retention times were compared to standard injections. Limit of quantitation (LOQ) was set at 3x baseline detection.

## (p)ppGpp detection and thin layer chromatography

*B. burgdorferi* cultures were grown at 35 °C in BSK + RS to late log phase, washed three times in cold dPBS and resuspended in BSK + RS at $5 \times 10^6$ Bb/ml containing 20 μCi/ml $^{32}$P either with or without IPTG as described and grown at 35 °C. At the times indicated, cells were collected by centrifugation at 6000 x g for 15 m at 4 °C, the supernatant discarded, and the cell pellets weighed. Nucleic acids were extracted by adding 20 μl of ice-cold 6.4 M formic acid and samples stored at -80 °C. Load volumes were standardized by pellet weight before separation on polyethylenimine (PEI) cellulose plates (EMD) by TLC as previously described[9].

## Statistics and reproducibility

All statistical analyses were performed using Prism software (v9.5.1; GraphPad Software, Inc.). All data is represented as mean ± SEM. Statistical analysis of growth curves and flow cytometry data was performed using two-way ANOVA. *p* values for cell length, qPCR DNA burdens, viability plating, IPTG detection and heavy amino acid labeling were determined for pairwise comparisons using two-tailed unpaired *t* tests. Exact *p*-values are provided when available, *p*-value < 0.05 was considered statistically significant.

## Reporting summary

Further information on research design is available in the Nature Portfolio Reporting Summary linked to this article.

## Data availability

All data supporting the findings of this study are available within the paper and its Supplementary Information. The source values underlying metabolomics data can be found in supplementary data 2, and for Figs. 2-4 and S4–5 can be found in supplementary data 4, while the original, uncropped blots are shown in Fig. S8. RNAseq files were deposited in the NCBI Sequence Read Archive under the following BioSample Accession numbers: SAMN53393123, SAMN53393124, SAMN53393125, SAMN53393126, SAMN53393127, SAMN53393128, SAMN53393129, SAMN53393130, SAMN53393131, SAMN53393132, SAMN53393133, SAMN53393134, SAMN53393135, SAMN53393136, SAMN53393137, SAMN53393138.

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

## Acknowledgements

We thank Rose Perry-Gottschalk, Alexander Stewart, Austin Athman, and Anita Mora of Visual and Medical Arts, NIAID Research Technology Branch for generation of overview Fig. This work was supported by the Division of Intramural Research/NIAID/NIH (AMG) and by the National Institutes of Health grants R01 AI130247 (DSS and DD) and U01 AI169840 (C. Davies, R.T. Marconi and DSS).

## Author contributions

A.M.G. designed research. A.K. and A.M.G. performed experimentation. A.M.G. performed general data analysis and statistics. E.B. and B.S. developed, performed and analyzed metabolomics assays. A.B.C. and L.M.M. performed and analyzed flow cytometry. D.D. performed and analyzed (p)ppGpp measurements. A.K., B.S., D.D., and A.M.G. wrote methods. A.M.G. drafted the manuscript. A.M.G., S.S., D.D., L.M.M., A.B.C., B.S., E.B., A.K. reviewed and edited the manuscript. This research was supported in part by the Intramural Research Program of the National Institutes of Health (NIH). The contributions of the NIH author(s) were made as part of their official duties as NIH federal staff, are in compliance with agency policy requirements, and are considered Works of the United States Government. However, the findings and conclusions presented in this paper are those of the authors and do not necessarily reflect the views of the NIH or the U.S. Department of Health and Human Services.

## Funding

## Competing interests

The authors declare no competing interests.
