## [Transparent Peer Review file · Communications Biology]

Dissection of amino acid acquisition pathways in *Borrelia burgdorferi* uncovers unique physiological responses

Corresponding Author: Dr Ashley Groshong

Version 0:

Reviewer comments:

Reviewer #1

(Remarks to the Author)

In this work, Kataria et al. defined the functions of BB0401 from Lyme disease bacteria as a transporter to acquire amino acids, glutamate and aspartate. The strength of this work is to take advantages of the authors' previously developed inducible platform to examine the in vitro and in vivo phenotypes of BB0401 mutants. The other strength is that the authors are able to compare their previously identified transporter OPP with BB0401 regarding their roles in conferring Lyme borreliosis enzootic cycle. The manuscript was well written and easy to understand. I only have few suggestions that may need the authors to address:

1. The authors generated an unrooted tree of several amino acid transporting genes from different bacteria and bb0401. The authors then concluded that "the putative *B. burgdorferi* GltP (BB0401), is more divergent but still groups within the Glt family" (Line 122 to 123). In order to be more concisely describe the BB0401 in the same group as other proteins in Glt family, it may be better to generate a rooted tree and show the confidence value that supports the branching point immediately before BB0401 diversifies.
2. In Fig. 3e and f, the *B. burgdorferi* burdens in the bb0401cond+IPTG and oppcond+IPTG were significantly reduced and not detectable, respectively, after nymph feeding. The authors then concluded these results as "bb0401 and opp are required for proliferation within the feeding nymphs and transmission. (line 297 to 298)" However, if bb0401 and opp indeed play a role for bacterial proliferation within feeding nymphs, the *B. burgdorferi* burdens in the bb0401cond+IPTG and oppcond+IPTG would remain the same before and after nymph feeding. The fact that the burdens of those strains were reduced/undetectable in Fig. 3e and 3f suggest that bb0401 and opp promote *B. burgdorferi* survival in the tick blood meal. Thus, the authors may need to revisit the paragraph from 297 to 329 and discuss the implication of such a function of bb0401 and opp to facilitate *B. burgdorferi* survival in fed nymphs.
3. It may be better to include a control strain (e.g., WT) in Fig. 2e so that the readers could compare the differences of lengths between WT and bb0401cond, and when the differences of the lengths begin to be significantly different between WT and bb0401cond.
4. The sentence from 268 to 271 starting from "While..." to "...not ear" is not clear.

Reviewer #2

(Remarks to the Author)

The current study by Kataria et al. elucidated how specific outer membrane transporters impact *B. burgdorferi* viability in vitro and throughout its enzootic life cycle, using separate conditional mutants of oligopeptide transport system (Opp) and bb0401, both regulated by exogenous addition of IPTG. They also provided in silico and compelling experimental evidence that BB0401 protein is a glutamate and aspartate transporter. Those findings stem from a series of well-designed experiments which certainly enrich our understanding of *Borrelia* nutritional biology and survivability through its tick-mouse infectious cycle. However, the experimental approaches require further authentication through a set of additional experiments, as indicated below:

Major Comment

1. Authors should verify their main experimental system of conditional mutagenesis, showing that the in vitro expression pattern of the targeted genes and proteins (Opp, BB0401), is directly regulated by varying concentrations of IPTG. In addition, the expression of the genes as well as the production of the proteins (if possible), in the presence or absence of IPTG, needs to be validated in mice and in ticks.
2. Fig. 3: The data shows three group/panels for Western blotting but only two for qPCR, which is confusing. For example, the group "bb0401cond" has no immune response in Fig 3a and is culture negative (meaning absence of infection), while the same group has spirochete levels similar to WT group in Fig 3c. Authors should explain these findings and include in the qPCR panels (a, d, l and j) all three groups (WT, bb0401cond in presence and in absence of IPTG).

Minor Comments

1. To better represent the evolutionary relationship of BB0401 with other glutamate and cystine transporters, authors should generate a more robust phylogenetic tree including more related transporters from other species. Note that there are many *B. burgdorferi* proteins which show similarities with proteins from other distant species - even from eukaryotes, especially in their binding domain or other regions. The gene accession number used for silico analysis should be included in the supplementary material.
2. The legends are confusing in places. For example, the legend for Fig 3 omits mention of the panels like g, h, l, and j. Similarly, the legend for Fig 4 (d-f) indicates "5 days" of starvation, while in the result section it is mentioned that the starvation period was "7 days". Authors should make sure that every legend in the manuscript contains clear information.
3. Fig S2 a, b: The labels of the graphs are difficult due to poor resolution.
4. Authors should avoid the use of acronyms, for example when they are referring to the transporters for the first time (especially in the abstract).

Reviewer #3

(Remarks to the Author)

This manuscript presents a detailed investigation into amino acid and peptide starvation responses in *Borrelia burgdorferi*, the causative agent of Lyme disease, with a specific focus on the functional characterization of a putative glutamate/aspartate transporter, GltP (BB0401). Building upon prior findings that peptide transport is essential for *B. burgdorferi* viability, the authors aim to delineate the physiological and metabolic consequences of targeted amino acid deprivation (via BB0401 disruption) in comparison to broad-spectrum peptide starvation (via the Opp system). Overall, the study offers a comprehensive examination of amino acid transport and starvation response mechanisms in *B. burgdorferi*. The manuscript is well written but would benefit from improvements in some areas of clarity and technical detail.

Minor comments:

1. Line 49. Please add a couple of sentences introducing the (p)ppGpp signaling molecule, including its function in bacterial stress responses. Additionally, since this is the first mention of DksA, I recommend including its full protein name and corresponding locus tag for clarity.
2. Line 81. Please revise to: "...with solved crystal structures..."
3. Line 92. Please specify the modeling tool used. For example: "Structurally, Swiss-Model (ref) generated BB0401 models using *P. horikoshii* GltPh as a template..."
4. Line 92. Delete the word "against" from the sentence for clarity.
5. Results section Lines 110-157. While Swiss-Model can be effective when using high-resolution templates and high sequence similarity, this is not the case for PDB entry 2nwl (2.96 Å resolution) and the BB0401-GltPh alignment, which shows only approx. 22% sequence similarity. Nevertheless, a quick alignment analysis shows that the AlphaFold-predicted structure of BB0401 aligns well with the Swiss-Model output you generated, particularly in the active site region discussed in the manuscript. So in this case, this approach you used has worked well. If AlphaFold was used, you might also consider evaluating trimer formation using AlphaFold-Multimer, as observed in GltPh (Lines 128-130).
6. Line 128. Add: (PDB ID 2nwl)
7. Line 132. Please consider the fact that some residues in the GltPh crystal structure (PDB ID 2nwl) interact with aspartate via their main-chain carbonyl groups, rather than side chains (e.g., Arg276, Val355). This implies that conservation of residue conformation, not just identity, is important for functional interpretation.
8. Figure 1. You are using the crystal structure of GltPh (PDB ID 2nwl), which was solved in complex with aspartate (Asp) but without sodium (Na), to illustrate the residues involved in Asp and Na binding, and to compare these residues with those in the predicted BB0401 model. Therefore, the figure legend should clearly state that the crystal structure of GltPh in complex with Asp (PDB ID 2nwl) is shown to highlight the residues involved in Asp binding (see Line 999).
9. Figure 1 panels c-f should be improved for clarity.. For example, in figure 1c provide a zoomed-in view, and ideally, illustrate hydrogen/ionic bonds as dotted lines. Remove non-relevant residues to improve clarity.
10. Figure 1d: Na is actually not present in the crystal structure of PDB entry 2nwl. Since Na is not present in 2nwl, consider using PDB ID 2nwx for panel 1d, which includes Na, instead.
11. Fig 1e: The legend states that predicted Glu-binding residues in GltPEc are shown, but the panel appears to display GltPh. Please clarify the structure used and ensure consistency.
12. Line 136: It is unclear whether docking was performed by the authors or cited from previous work. Clarify whether the residues shown are from experimental data or docking predictions, and explain the rationale for their mapping onto GltPh.
13. Line 151-157. You state that SwissDock predicted BB0401-Asp/Glu models that aligned well with the GltPh-Asp complex, but no structural overlay is shown in Figure 1f. Please provide such an overlay and improve figure resolution and labeling, as the current panel lacks sufficient detail for evaluation.

14. Line 141. Based on the crystal structure of GltPh (PDB ID 2nwl) ...
15. Line 141. GltPEc should be mentioned. And for the Swiss-Model predictions, please include the GMQE and/or QMEANDisCo values to support model quality.
16. Line 143-145. Please include units for RMSD values and provide separate RMSDs for GltPEc–GltPh and BB0401–GltPh comparisons.
17. Line 147. Please revise to: ... pockets showed limited conservation ..
18. Line 569, 571, 574. Where possible, decipher abbreviations upon first use. Also, cite the software tools used: MUSCLE, PhyML, iTOL, ChimeraX, and SwissDock.
19. Line 574. Add: (PDB ID 2nwl).
20. Line 652. Instead of "0 mM", please use phrasing such as: "in the absence of added .." or "without supplementing..."
21. Materials and Methods: Please maintain consistency in volume unit formatting throughout the section. For example, choose either μl or μL , and be consistent with capitalization across ml, l, and L.

Version 1:

Reviewer comments:

Reviewer #1

(Remarks to the Author)

I think the authors did a great job in revising the manuscript, and I don't have any further questions/comments

Reviewer #2

(Remarks to the Author)

Authors sufficiently responded to most of the requested modifications. I have minor comments on the new set of experimental data (Fig 2a and Fig S5c), as described below:

Fig.2a: Authors should address the following :the DNA size of bb0401 for positive control (marked as+) is higher than the one observed for wt and bb0401cond induced by the IPTG. What has been used as positive control and why the band is of higher MW?

Fig.S5c: The graph represents the transcripts of wt and bb0401cond group, but the legend indicates wt and bb0401cond plus IPTG. Authors should correct this inconsistency. Please check additional typos elsewhere. In addition, does the legend in y-axis denote actual copies or relative copies?

Line 213: please correct the typo 0.1mM to 0.01M IPTG. Please check additional typos elsewhere.

Reviewer #3

(Remarks to the Author)

All my previous concerns have been addressed in the revised version. I am satisfied with the authors' revisions and have no further comments.

We appreciate the reviewers' responses and have done our best to address all points. We apologize for the lengthy revision timeline and for anything we could not experimentally address in this revision. The current situation within the federal research agencies limited some experimentation.

Reviewer #1	Author
Major Comments	
1. The authors generated an unrooted tree of several amino acid transporting genes from different bacteria and bb0401. The authors then concluded that “the putative B. burgdorferi GltP (BB0401), is more divergent but still groups within the Glt family” (Line 122 to 123). In order to be more concisely describe the BB0401 in the same group as other proteins in Glt family, it may be better to generate a rooted tree and show the confidence value that supports the branching point immediately before BB0401 diversifies.	We have modified the tree as requested. We have also added additional homologs as requested by Reviewer 2. See Fig S1 and Table S3 for updated tree and legend information and lines 149-169 for modified manuscript language.
2. In Fig. 3e and f, the B. burgdorferi burdens in the bb0401cond+IPTG and oppcond+IPTG were significantly reduced and not detectable, respectively, after nymph feeding. The authors then concluded these results as “ bb0401 and opp are required for proliferation within the feeding nymphs and transmission. (line 297 to 298)” However, if bb0401 and opp indeed play a role for bacterial proliferation within feeding nymphs, the B. burgdorferi burdens in the bb0401cond+IPTG and oppcond+IPTG would remain the same before and after nymph feeding. The fact that the burdens of those strains were reduced/undetectable in Fig. 3e and 3f suggest that bb0401 and opp promote B. burgdorferi survival in the tick blood meal. Thus, the authors may need to revisit the paragraph from 297 to 329 and discuss the implication of such a function of bb0401 and opp to facilitate B. burgdorferi survival in fed nymphs.	We appreciate the reviewer’s discrimination between the spirochetes’ failure to proliferate compared to the overall viability within the feeding tick. We have modified the title of this section (line 503), clarified the phenotype within the section (line 522), and noted this difference in the results (lines 749-752).

3. It may be better to include a control strain (e.g., WT) in Fig. 2e so that the readers could compare the differences of lengths between WT and bb0401cond, and when the differences of the lengths begin to be significantly different between WT and bb0401cond.	We have added measurements of wild-type bacteria over the same time frame in Fig 2f for comparison purposes.
4. The sentence from 268 to 271 starting from “While...” to “...not ear” is not clear.	We have modified the statement (lines 461-478) to clarify. A previous study had used the in vivo IPTG complementation method to show expression of OspC could be restored via the inducible system. However, because OspC is responsible for establishing early infection, the study only evaluated mice 1 week post infection. The inability to culture spirochetes from all tissues also suggested that there could be a problem with penetration of IPTG via this route to all tissues. We were able to show IPTG-induction to be functional for a protracted amount of time and penetration to all tissues that we use to assess dissemination based on culture results.
Reviewer #2	Author
Major Comments	
1. Authors should verify their main experimental system of conditional mutagenesis, showing that the in vitro expression pattern of the targeted genes and proteins (Opp, BB0401), is directly regulated by varying concentrations of IPTG. In addition, the expression of the genes as well as the production of the proteins (if possible), in the presence or absence of IPTG, needs to be validated in mice and in ticks.	While this conditional mutagenesis system has been validated in vitro by us and others in earlier studies, we, to the best of our current ability to perform experiments available reagents, have included an assessment of expression for bb0401. We did attempt, multiple times, to utilize antibody-based protein detection for this target. We have included in Fig 2b and lines 330-344 a description of transcript quantification for bb0401 in vitro. We were stymied in utilizing qRT-PCR for these studies as it appeared there was RNA degradation under starvation conditions (0.01 and 0mM IPTG), therefore we had to lean on RT-PCR to achieve a semi-quantitative analysis of the mutant. We believe these results are consistent with results we showed in Groshong et al. 2012 where

	we generated a conditional mutant of rrp2 as these mutants were generated using the same inducible vector and were able to show via Western blot that protein expression was titratable with different IPTG concentrations. Additionally, the growth phenotype was clearly titratable for both opp^{cond} (Groshong et al. 2017) and bb0401^{cond} further supporting expression differences. Transcript quantification in vivo for B. burgdorferi can be quite difficult due to low burdens of the bacteria in tissues; however, we were able to quantify bb0401 transcripts in infected hearts that we had from the 4 wk infection timepoint (Fig S5c, lines 457-460), all other tissues were below the limit of detection for bb0401. All tick samples had been processed for DNA so we had no samples to test for RNA. Due to limitations in acquiring reagents in these challenging times at federal scientific agencies, we were unable to quantify the oppDF transcripts for this study. We did attempt to express recombinant protein; unfortunately, we were unsuccessful. While we could not obtain a qRT-PCR assay for oppDF at this time, we expect the overall RNA degradation we showed for the opp^{cond} during starvation (Fig S7) would result in similar hurdles to using qRT-PCR for quantitation as seen with bb0401. We do not know at this point whether this RNA degradation phenotype is unique to amino acid starvation conditions, but our intention is to tackle these questions in our subsequent manuscript.
2. Fig. 3: The data shows three group/panels for Western blotting but only two for qPCR, which is confusing. For example, the group “bb0401cond” has no immune response in Fig 3a and is culture negative (meaning absence of infection), while the same group has spirochete	We apologize for the confusion. We did not originally perform qPCR on the negative culture tissues which made direct comparison across panels difficult. We have now included the qPCR for all tissues (Fig 3c,d,i,j) and modified the related text (line 453-454). The mice that

levels similar to WT group in Fig 3c. Authors should explain these findings and include in the qPCR panels (a, d, I and j) all three groups (WT, bb0401cond in presence and in absence of IPTG).	did not receive IPTG were culture negative, seronegative, and negative by qPCR.
Minor Comments	
1. To better represent the evolutionary relationship of BB0401 with other glutamate and cystine transporters, authors should generate a more robust phylogenetic tree including more related transporters from other species. Note that there are many B. burgdorferi proteins which show similarities with proteins from other distant species - even from eukaryotes, especially in their binding domain or other regions. The gene accession number used for silico analysis should be included in the supplementary material.	We have modified the tree as requested. We have also changed the tree to a rooted tree as requested by Reviewer 1. See Fig S1 and Table S3 for updated tree and legend information and lines 149-169 for modified manuscript language.
2. The legends are confusing in places. For example, the legend for Fig 3 omits mention of the panels like g, h, I, and j. Similarly, the legend for Fig 4 (d-f) indicates "5 days" of starvation, while in the result section it is mentioned that the starvation period was "7 days". Authors should make sure that every legend in the manuscript contains clear information.	We apologize for the confusion and have corrected legends for Fig 3 and Fig 4.
3. Fig S2 a, b: The labels of the graphs are difficult due to poor resolution.	We have relabeled figure panels a and b (now Fig S3) to improve quality.
4. Authors should avoid the use of acronyms, for example when they are referring to the transporters for the first time (especially in the abstract).	We have removed the acronyms from the abstract and reviewed the remaining manuscript for acronyms which were not defined upon first use.
Reviewer #3	Author
1. Line 49. Please add a couple of sentences introducing the (p)ppGpp signaling molecule, including its function in bacterial stress responses. Additionally, since this is the first mention of DksA, I recommend including its full protein name and corresponding locus tag for clarity.	We have expanded on the functionality of (p)ppGpp, Rel, and DksA (lines 64-72). To maintain consistency throughout we have added in full protein names (when available) and locus for all genes upon first mention.
2. Line 81. Please revise to: "...with solved crystal structures..."	The requested revision has been completed, see line 113

3. Line 92. Please specify the modeling tool used. For example: "Structurally, Swiss-Model (ref) generated BB0401 models using P. horikoshii GltPh as a template..."	The requested revision has been completed, see line 124-125. As we had previously introduced the P. horikoshii Glt_{PH} as "Glt_{PH}" on line 113, we used this annotation and included the PDB designation at this point in the manuscript.
4. Line 92. Delete the word "against" from the sentence for clarity.	The requested revision has been completed, see line 125
5. Results section Lines 110-157. While Swiss-Model can be effective when using high-resolution templates and high sequence similarity, this is not the case for PDB entry 2nwl (2.96 Å resolution) and the BB0401-GltPh alignment, which shows only approx. 22% sequence similarity. Nevertheless, a quick alignment analysis shows that the AlphaFold-predicted structure of BB0401 aligns well with the Swiss-Model output you generated, particularly in the active site region discussed in the manuscript. So in this case, this approach you used has worked well. If AlphaFold was used, you might also consider evaluating trimer formation using AlphaFold-Multimer, as observed in GltPh (Lines 128–130).	We appreciate the reviewer's insight into the use of template modeling. We did indeed model with AlphaFold2-Multimer given the low similarity between BB0401 and 2nwl but had proceeded with the SwissModel version due to high similarity between the two models. Given the reviewer's comments, we have included the AlphaFold2-Multimer model aligned with our SwissModel BB0401 in what is now Fig S2 and updated language in the results section in lines 190-194 to include this data within the context that the reviewer recommended and to support our findings related to the models.
6. Line 128. Add: (PDB ID 2nwl)	The requested revision has been completed, see line 170
7. Line 132. Please consider the fact that some residues in the GltPh crystal structure (PDB ID 2nwl) interact with aspartate via their main-chain carbonyl groups, rather than side chains (e.g., Arg276, Val355). This implies that conservation of residue conformation, not just identity, is important for functional interpretation.	We appreciate the Reviewer's additional insight to the binding mechanisms of Glt_{PH} and recognize that conservation of main-chain interactions versus side chain interactions are worth evaluating separately. We have added context for these subtleties in the manuscript text in lines 174-176 and 249-250, and we have amended the liganding residues table in Fig 1 to denote residues which interact via main-chain interactions.
8. Figure 1. You are using the crystal structure of GltPh (PDB ID 2nwl), which was solved in complex with aspartate (Asp) but without sodium (Na), to	We have updated the figure legend for clarity to note that 2nwl is complexed with Asp, the glutamate site is based on predictions by Rahman et al, and utilized

illustrate the residues involved in Asp and Na binding, and to compare these residues with those in the predicted BB0401 model. Therefore, the figure legend should clearly state that the crystal structure of GltPh in complex with Asp (PDB ID 2nwl) is shown to highlight the residues involved in Asp binding (see Line 999).	2nwx for the residue alignment of sodium binding sites and Na.
9. Figure 1 panels c-f should be improved for clarity.. For example, in figure 1c provide a zoomed-in view, and ideally, illustrate hydrogen/ionic bonds as dotted lines. Remove non-relevant residues to improve clarity.	We have updated the panels in Fig 1 binding site panels. Panels now show relevant residues ligands and bonds where possible to improve clarity. We have also added residue labels to aid in orientation.
10. Figure 1d: Na is actually not present in the crystal structure of PDB entry 2nwl. Since Na is not present in 2nwl, consider using PDB ID 2nwx for panel 1d, which includes Na, instead.	We have replaced the Glt_{Ph} model in Fig panel 1d with 2nwx so that we can demonstrate both Asp and Na binding pockets.
11. Fig 1e: The legend states that predicted Glu-binding residues in GltPEc are shown, but the panel appears to display GltPh. Please clarify the structure used and ensure consistency.	The legend has been updated to clarify the details of each panel. Panel 1d has specifically been updated to show both the asp binding (as defined by 2nwl) and corresponding glu binding residues as determined by AutoDock in Rahman et al. via the 2nwl structure for consistency.
12. Line 136: It is unclear whether docking was performed by the authors or cited from previous work. Clarify whether the residues shown are from experimental data or docking predictions, and explain the rationale for their mapping onto GltPh.	We have clarified in the manuscript text that the GltPEc docking studies were performed by authors from a previous study (lines 182-186). Our use of docking studies were limited to evaluation of the BB0401 model as described in lines 254-263. We have also clarified this detail in the Fig 1 legend.
13. Line 151-157. You state that SwissDock predicted BB0401-Asp/Glu models that aligned well with the GltPh-Asp complex, but no structural overlay is shown in Figure 1f. Please provide such an overlay and improve figure resolution and labeling, as the current panel lacks sufficient detail for evaluation.	We have updated Fig 1 to show an overlay of GltPh asp binding site from 2nwl with the SwissDock results for both Asp and Glu binding to BB0401 (panel e), as well as individual panels for BB0401 binding to Asp (panel f) and Glu (panel g) to provide the reader with clearer visualizations of the structures. We have additionally modified the manuscript to note common liganding residues between

	the docking study and the GltPh/GltPEc studies in lines 257-261.
14. Line 141. Based on the crystal structure of GltPh (PDB ID 2nwl) ...	The requested revision has been completed, see line 188.
15. Line 141. GltPEc should be mentioned. And for the Swiss-Model predictions, please include the GMQE and/or QMEANDisCo values to support model quality.	The requested data has been added, see line 187-190.
16. Line 143-145. Please include units for RMSD values and provide separate RMSDs for GltPEc–GltPh and BB0401–GltPh comparisons.	The requested data has been added, see lines 193 and 245.
17. Line 147. Please revise to: .. pockets showed limited conservation ..	The requested revision has been completed, see line 248.
18. Line 569, 571, 574. Where possible, decipher abbreviations upon first use. Also, cite the software tools used: MUSCLE, PhyML, iTOL, ChimeraX, and SwissDock.	The requested revision has been completed, see lines 824-837 and citations have been provided throughout.
19. Line 574. Add: (PDB ID 2nwl).	The requested revision has been completed, see line 831.
20. Line 652. Instead of "0 mM", please use phrasing such as: "in the absence of added .."or "without supplementing..."	The requested revision has been completed, see line 933.
21. Materials and Methods: Please maintain consistency in volume unit formatting throughout the section. For example, choose either μl or μL , and be consistent with capitalization across ml, l, and L.	We have reviewed all unit measurements for consistent formatting throughout.

Reviewer 2 Comment	Author Response
Fig.2a: Authors should address the following :the DNA size of bb0401 for positive control (marked as+) is higher than the one observed for wt and bb0401cond induced by the IPTG. What has been used as positive control and why the band is of higher MW?	We have amended the legend for Fig 2a to note that the + ctrl is genomic DNA (gDNA). We also noticed this size discrepancy when visualizing the samples. We were able to purify the bands from the PCR amplifications and sequenced the PCR product. We found no deletions/insertions that would account for a change in size of the transcript compared with the gDNA. We did have to load significantly more reaction for the RNA reactions compared with the gDNA reaction (10x dilution of the gDNA sample and 15x more RNA sample) to get the loading within relative range for visualization. This would result in some differences between the positive control and RNA sample composition such as carryover from cDNA reagents as well as dramatically different load volumes. Given we were able to confirm sequence and the wt RNA amplified at the same size as the bb0401cond we believe the size discrepancy relates to these elements as we have seen these effects in loading volumes in PCR before. We have added language to the results section that notes the discrepancy, our sequencing results, and our possible explanation for the difference. (Line 216-222)
Fig.S5c: The graph represents the transcripts of wt and bb0401cond group, but the legend indicates wt and bb0401cond plus IPTG. Authors should correct this inconsistency. Please check additional typos elsewhere. In addition, does the legend in y-axis denote actual copies or relative copies?	We have corrected the sample name to bb0401cond+IPTG for the graph in Fig S5c. The y-axis denotes actual quantification of bb0401 per 100 copies of flaB as denoted on the y-axis. We have added this detail in the legend for clarification.
Line 213: please correct the typo 0.1mM to 0.01M IPTG. Please check additional typos elsewhere.	This correction has been made.

We appreciate the reviewers' responses and have done our best to address all points. We apologize for the lengthy revision timeline and for anything we could not experimentally address in this revision. The current situation within the federal research agencies limited some experimentation.

Reviewer #1	Author
Major Comments	
1. The authors generated an unrooted tree of several amino acid transporting genes from different bacteria and bb0401. The authors then concluded that “the putative B. burgdorferi GltP (BB0401), is more divergent but still groups within the Glt family” (Line 122 to 123). In order to be more concisely describe the BB0401 in the same group as other proteins in Glt family, it may be better to generate a rooted tree and show the confidence value that supports the branching point immediately before BB0401 diversifies.	We have modified the tree as requested. We have also added additional homologs as requested by Reviewer 2. See Fig S1 and Table S3 for updated tree and legend information and lines 149-169 for modified manuscript language.
2. In Fig. 3e and f, the B. burgdorferi burdens in the bb0401cond+IPTG and oppcond+IPTG were significantly reduced and not detectable, respectively, after nymph feeding. The authors then concluded these results as “ bb0401 and opp are required for proliferation within the feeding nymphs and transmission. (line 297 to 298)” However, if bb0401 and opp indeed play a role for bacterial proliferation within feeding nymphs, the B. burgdorferi burdens in the bb0401cond+IPTG and oppcond+IPTG would remain the same before and after nymph feeding. The fact that the burdens of those strains were reduced/undetectable in Fig. 3e and 3f suggest that bb0401 and opp promote B. burgdorferi survival in the tick blood meal. Thus, the authors may need to revisit the paragraph from 297 to 329 and discuss the implication of such a function of bb0401 and opp to facilitate B. burgdorferi survival in fed nymphs.	We appreciate the reviewer’s discrimination between the spirochetes’ failure to proliferate compared to the overall viability within the feeding tick. We have modified the title of this section (line 503), clarified the phenotype within the section (line 522), and noted this difference in the results (lines 749-752).

3. It may be better to include a control strain (e.g., WT) in Fig. 2e so that the readers could compare the differences of lengths between WT and bb0401cond, and when the differences of the lengths begin to be significantly different between WT and bb0401cond.	We have added measurements of wild-type bacteria over the same time frame in Fig 2f for comparison purposes.
4. The sentence from 268 to 271 starting from “While...” to “...not ear” is not clear.	We have modified the statement (lines 461-478) to clarify. A previous study had used the in vivo IPTG complementation method to show expression of OspC could be restored via the inducible system. However, because OspC is responsible for establishing early infection, the study only evaluated mice 1 week post infection. The inability to culture spirochetes from all tissues also suggested that there could be a problem with penetration of IPTG via this route to all tissues. We were able to show IPTG-induction to be functional for a protracted amount of time and penetration to all tissues that we use to assess dissemination based on culture results.
Reviewer #2	Author
Major Comments	
1. Authors should verify their main experimental system of conditional mutagenesis, showing that the in vitro expression pattern of the targeted genes and proteins (Opp, BB0401), is directly regulated by varying concentrations of IPTG. In addition, the expression of the genes as well as the production of the proteins (if possible), in the presence or absence of IPTG, needs to be validated in mice and in ticks.	While this conditional mutagenesis system has been validated in vitro by us and others in earlier studies, we, to the best of our current ability to perform experiments available reagents, have included an assessment of expression for bb0401. We did attempt, multiple times, to utilize antibody-based protein detection for this target. We have included in Fig 2b and lines 330-344 a description of transcript quantification for bb0401 in vitro. We were stymied in utilizing qRT-PCR for these studies as it appeared there was RNA degradation under starvation conditions (0.01 and 0mM IPTG), therefore we had to lean on RT-PCR to achieve a semi-quantitative analysis of the mutant. We believe these results are consistent with results we showed in Groshong et al. 2012 where

	we generated a conditional mutant of rrp2 as these mutants were generated using the same inducible vector and were able to show via Western blot that protein expression was titratable with different IPTG concentrations. Additionally, the growth phenotype was clearly titratable for both opp^{cond} (Groshong et al. 2017) and bb0401^{cond} further supporting expression differences. Transcript quantification in vivo for B. burgdorferi can be quite difficult due to low burdens of the bacteria in tissues; however, we were able to quantify bb0401 transcripts in infected hearts that we had from the 4 wk infection timepoint (Fig S5c, lines 457-460), all other tissues were below the limit of detection for bb0401. All tick samples had been processed for DNA so we had no samples to test for RNA. Due to limitations in acquiring reagents in these challenging times at federal scientific agencies, we were unable to quantify the oppDF transcripts for this study. We did attempt to express recombinant protein; unfortunately, we were unsuccessful. While we could not obtain a qRT-PCR assay for oppDF at this time, we expect the overall RNA degradation we showed for the opp^{cond} during starvation (Fig S7) would result in similar hurdles to using qRT-PCR for quantitation as seen with bb0401. We do not know at this point whether this RNA degradation phenotype is unique to amino acid starvation conditions, but our intention is to tackle these questions in our subsequent manuscript.
2. Fig. 3: The data shows three group/panels for Western blotting but only two for qPCR, which is confusing. For example, the group “bb0401cond” has no immune response in Fig 3a and is culture negative (meaning absence of infection), while the same group has spirochete	We apologize for the confusion. We did not originally perform qPCR on the negative culture tissues which made direct comparison across panels difficult. We have now included the qPCR for all tissues (Fig 3c,d,l,j) and modified the related text (line 453-454). The mice that

levels similar to WT group in Fig 3c. Authors should explain these findings and include in the qPCR panels (a, d, I and j) all three groups (WT, bb0401cond in presence and in absence of IPTG).	did not receive IPTG were culture negative, seronegative, and negative by qPCR.
Minor Comments	
1. To better represent the evolutionary relationship of BB0401 with other glutamate and cystine transporters, authors should generate a more robust phylogenetic tree including more related transporters from other species. Note that there are many B. burgdorferi proteins which show similarities with proteins from other distant species - even from eukaryotes, especially in their binding domain or other regions. The gene accession number used for silico analysis should be included in the supplementary material.	We have modified the tree as requested. We have also changed the tree to a rooted tree as requested by Reviewer 1. See Fig S1 and Table S3 for updated tree and legend information and lines 149-169 for modified manuscript language.
2. The legends are confusing in places. For example, the legend for Fig 3 omits mention of the panels like g, h, I, and j. Similarly, the legend for Fig 4 (d-f) indicates "5 days" of starvation, while in the result section it is mentioned that the starvation period was "7 days". Authors should make sure that every legend in the manuscript contains clear information.	We apologize for the confusion and have corrected legends for Fig 3 and Fig 4.
3. Fig S2 a, b: The labels of the graphs are difficult due to poor resolution.	We have relabeled figure panels a and b (now Fig S3) to improve quality.
4. Authors should avoid the use of acronyms, for example when they are referring to the transporters for the first time (especially in the abstract).	We have removed the acronyms from the abstract and reviewed the remaining manuscript for acronyms which were not defined upon first use.
Reviewer #3	Author
1. Line 49. Please add a couple of sentences introducing the (p)ppGpp signaling molecule, including its function in bacterial stress responses. Additionally, since this is the first mention of DksA, I recommend including its full protein name and corresponding locus tag for clarity.	We have expanded on the functionality of (p)ppGpp, Rel, and DksA (lines 64-72). To maintain consistency throughout we have added in full protein names (when available) and locus for all genes upon first mention.
2. Line 81. Please revise to: "...with solved crystal structures..."	The requested revision has been completed, see line 113

3. Line 92. Please specify the modeling tool used. For example: "Structurally, Swiss-Model (ref) generated BB0401 models using P. horikoshii GltPh as a template..."	The requested revision has been completed, see line 124-125. As we had previously introduced the P. horikoshii Glt_{Ph} as "Glt_{Ph}" on line 113, we used this annotation and included the PDB designation at this point in the manuscript.
4. Line 92. Delete the word "against" from the sentence for clarity.	The requested revision has been completed, see line 125
5. Results section Lines 110-157. While Swiss-Model can be effective when using high-resolution templates and high sequence similarity, this is not the case for PDB entry 2nwl (2.96 Å resolution) and the BB0401-GltPh alignment, which shows only approx. 22% sequence similarity. Nevertheless, a quick alignment analysis shows that the AlphaFold-predicted structure of BB0401 aligns well with the Swiss-Model output you generated, particularly in the active site region discussed in the manuscript. So in this case, this approach you used has worked well. If AlphaFold was used, you might also consider evaluating trimer formation using AlphaFold-Multimer, as observed in GltPh (Lines 128–130).	We appreciate the reviewer's insight into the use of template modeling. We did indeed model with AlphaFold2-Multimer given the low similarity between BB0401 and 2nwl but had proceeded with the SwissModel version due to high similarity between the two models. Given the reviewer's comments, we have included the AlphaFold2-Multimer model aligned with our SwissModel BB0401 in what is now Fig S2 and updated language in the results section in lines 190-194 to include this data within the context that the reviewer recommended and to support our findings related to the models.
6. Line 128. Add: (PDB ID 2nwl)	The requested revision has been completed, see line 170
7. Line 132. Please consider the fact that some residues in the GltPh crystal structure (PDB ID 2nwl) interact with aspartate via their main-chain carbonyl groups, rather than side chains (e.g., Arg276, Val355). This implies that conservation of residue conformation, not just identity, is important for functional interpretation.	We appreciate the Reviewer's additional insight to the binding mechanisms of Glt_{Ph} and recognize that conservation of main-chain interactions versus side chain interactions are worth evaluating separately. We have added context for these subtleties in the manuscript text in lines 174-176 and 249-250, and we have amended the liganding residues table in Fig 1 to denote residues which interact via main-chain interactions.
8. Figure 1. You are using the crystal structure of GltPh (PDB ID 2nwl), which was solved in complex with aspartate (Asp) but without sodium (Na), to	We have updated the figure legend for clarity to note that 2nwl is complexed with Asp, the glutamate site is based on predictions by Rahman et al, and utilized

illustrate the residues involved in Asp and Na binding, and to compare these residues with those in the predicted BB0401 model. Therefore, the figure legend should clearly state that the crystal structure of GltPh in complex with Asp (PDB ID 2nwl) is shown to highlight the residues involved in Asp binding (see Line 999).	2nwx for the residue alignment of sodium binding sites and Na.
9. Figure 1 panels c-f should be improved for clarity.. For example, in figure 1c provide a zoomed-in view, and ideally, illustrate hydrogen/ionic bonds as dotted lines. Remove non-relevant residues to improve clarity.	We have updated the panels in Fig 1 binding site panels. Panels now show relevant residues ligands and bonds where possible to improve clarity. We have also added residue labels to aid in orientation.
10. Figure 1d: Na is actually not present in the crystal structure of PDB entry 2nwl. Since Na is not present in 2nwl, consider using PDB ID 2nwx for panel 1d, which includes Na, instead.	We have replaced the Glt_{Ph} model in Fig panel 1d with 2nwx so that we can demonstrate both Asp and Na binding pockets.
11. Fig 1e: The legend states that predicted Glu-binding residues in GltPEc are shown, but the panel appears to display GltPh. Please clarify the structure used and ensure consistency.	The legend has been updated to clarify the details of each panel. Panel 1d has specifically been updated to show both the asp binding (as defined by 2nwl) and corresponding glu binding residues as determined by AutoDock in Rahman et al. via the 2nwl structure for consistency.
12. Line 136: It is unclear whether docking was performed by the authors or cited from previous work. Clarify whether the residues shown are from experimental data or docking predictions, and explain the rationale for their mapping onto GltPh.	We have clarified in the manuscript text that the GltPEc docking studies were performed by authors from a previous study (lines 182-186). Our use of docking studies were limited to evaluation of the BB0401 model as described in lines 254-263. We have also clarified this detail in the Fig 1 legend.
13. Line 151-157. You state that SwissDock predicted BB0401-Asp/Glu models that aligned well with the GltPh-Asp complex, but no structural overlay is shown in Figure 1f. Please provide such an overlay and improve figure resolution and labeling, as the current panel lacks sufficient detail for evaluation.	We have updated Fig 1 to show an overlay of GltPh asp binding site from 2nwl with the SwissDock results for both Asp and Glu binding to BB0401 (panel e), as well as individual panels for BB0401 binding to Asp (panel f) and Glu (panel g) to provide the reader with clearer visualizations of the structures. We have additionally modified the manuscript to note common liganding residues between

	the docking study and the GltPh/GltPEc studies in lines 257-261.
14. Line 141. Based on the crystal structure of GltPh (PDB ID 2nwl) ...	The requested revision has been completed, see line 188.
15. Line 141. GltPEc should be mentioned. And for the Swiss-Model predictions, please include the GMQE and/or QMEANDisCo values to support model quality.	The requested data has been added, see line 187-190.
16. Line 143-145. Please include units for RMSD values and provide separate RMSDs for GltPEc–GltPh and BB0401–GltPh comparisons.	The requested data has been added, see lines 193 and 245.
17. Line 147. Please revise to: .. pockets showed limited conservation ..	The requested revision has been completed, see line 248.
18. Line 569, 571, 574. Where possible, decipher abbreviations upon first use. Also, cite the software tools used: MUSCLE, PhyML, iTOL, ChimeraX, and SwissDock.	The requested revision has been completed, see lines 824-837 and citations have been provided throughout.
19. Line 574. Add: (PDB ID 2nwl).	The requested revision has been completed, see line 831.
20. Line 652. Instead of "0 mM", please use phrasing such as: "in the absence of added .."or "without supplementing..."	The requested revision has been completed, see line 933.
21. Materials and Methods: Please maintain consistency in volume unit formatting throughout the section. For example, choose either μl or μL , and be consistent with capitalization across ml, l, and L.	We have reviewed all unit measurements for consistent formatting throughout.